# Cross-Domain Offline Policy Adaptation via Selective Transition Correction

## Abstract

It remains a critical challenge to adapt policies across domains with mismatched dynamics in reinforcement learning (RL). In this paper, we study cross-domain offline RL, where an offline dataset from another similar source domain can be accessed to enhance policy learning upon a target domain dataset. Directly merging the two datasets may lead to suboptimal performance due to potential dynamics mismatches. Existing approaches typically mitigate this issue through source domain transition filtering or reward modification, which, however, may lead to insufficient exploitation of the valuable source domain data. Instead, we propose to modify the source domain data into the target domain data. To that end, we leverage an inverse policy model and a reward model to correct the actions and rewards of source transitions, explicitly achieving alignment with the target dynamics. Since limited data may result in inaccurate model training, we further employ a forward dynamics model to retain corrected samples that better match the target dynamics than the original transitions. Consequently, we propose the Selective Transition Correction (STC) algorithm, which enables reliable usage of source domain data for policy adaptation. Experiments on various environments with dynamics shifts demonstrate that STC achieves superior performance against existing baselines.

## 1 Introduction

Reinforcement learning (RL) typically requires extensive interactions to train effective policies for a new task, which can be costly or infeasible in real-world applications (Levine et al., 2020; Eysenbach et al., 2021; Torne et al., 2024). In contrast, humans can rapidly adapt to new but structurally similar tasks once they have mastered a related skill (Lyu et al., 2025). This motivates the design of RL agents capable of leveraging experience from a similar domain (e.g., a simulator) to enhance learning efficiency in the target domain, a setting commonly referred to as the policy adaptation problem (Xu et al., 2023; Lyu et al., 2024b). A key challenge in this setting is the potential dynamics mismatch between the source domain and the target domain, which can significantly degrade the performance of the policy.

There are many researches focusing on the online policy adaptation setting where either the source or target domain is online. They fulfill policy adaptation by training domain classifiers (Eysenbach et al., 2021), filtering data via value difference (Xu et al., 2023), capturing representation mismatch (Lyu et al., 2024a), etc. However, in many real-world scenarios, online interactions can be costly or even unsafe. This motivates a shift of focus to the *offline policy adaptation* problem, or cross-domain offline RL (Wen et al., 2024; Lyu et al., 2025), where both the source domain and the target domain are offline. Existing cross-domain offline RL methods include filtering source domain data based on mutual information (Wen et al., 2024) or optimal transport (Lyu et al., 2025), augmenting source transition rewards by training domain classifiers (Liu et al., 2022), etc. These approaches typically mitigate the dynamics mismatch between datasets from two domains by selecting source transitions that resemble target domain data or penalizing dissimilar ones. Most existing methods still run on relatively large target-domain datasets. In practice, however, target-domain offline data are often expensive to collect and therefore limited. Moreover, when abundant target-domain data are accessible, the necessity of source domain transitions is substantially diminished. We instead consider the more challenging and practically relevant setting of offline policy adaptation with very limited target-domain

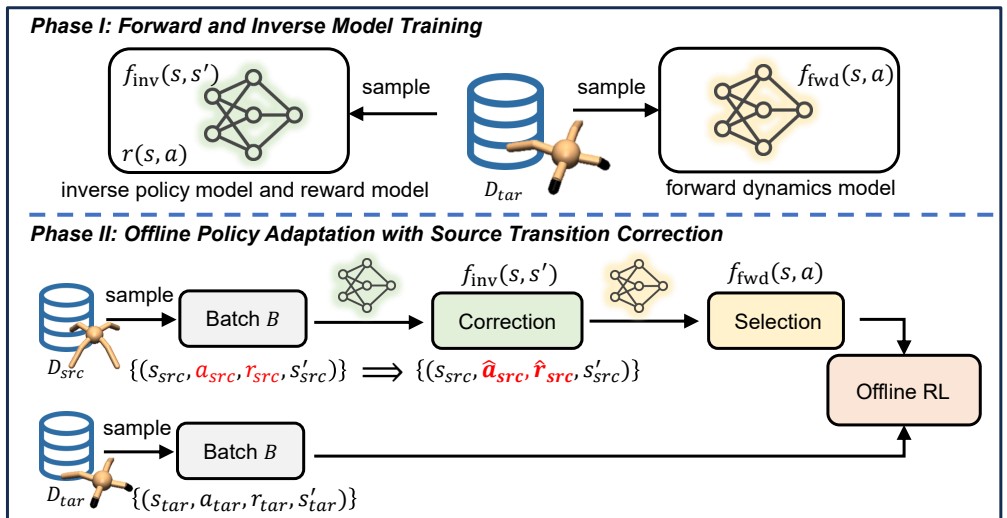

Figure 1: **Training pipeline of our proposed STC algorithm.** In ***Phase I***, we train the forward dynamics model $f_{\text{fwd}}(s, a)$, the reward model $r(s, a)$, and the inverse policy model $f_{\text{inv}}(s, s')$. These models are trained to capture the bidirectional dynamics transition information in the target domain dataset. In ***Phase II***, we sample data from $D_{\text{src}}$ and $D_{\text{tar}}$ to train an offline RL agent, where we correct the actions and rewards in the source domain transition tuple by using the inverse policy model. We further use the forward dynamics model to selectively correct source transitions to better align with the target domain.

data, consistent with OTDF (Lyu et al., 2025), where single-domain methods perform poorly. Under this setting, the limitations of existing cross-domain methods become more evident. By retaining only a subset of source transitions or downweighting mismatched ones, these methods may discard potentially useful source samples and result in inefficient use of the source dataset.

To mitigate this issue, we propose to *modify source transitions into target domain data* such that more source transitions can align with the target domain dataset, even those exhibiting substantial dynamics discrepancies. We leverage the inverse policy model trained on the target dataset to correct actions and rewards in the source domain dataset, which predicts the action label given the current state and the next state. We utilize the inverse policy model to replace the original source actions with ones that are more consistent with target dynamics. Based on the revised action, we approximately adjust the reward label using Taylor expansion and a trained reward model. We theoretically analyze the dynamics discrepancy of the corrected data against the target dataset and the value discrepancy on the corrected data and the source dataset. We further show the performance bound on the corrected data and the true target domain, which can be tighter if the source transitions are well corrected.

However, in practice, the target domain dataset is limited, which inevitably introduces approximation errors in the inverse policy model. Blindly correcting all source transitions may lead to performance degradation, as inaccurate predictions can produce poor corrected source transitions. To address this issue, we train a forward dynamics model based on the target domain dataset and introduce a selection mechanism that selectively corrects source dataset samples to achieve better alignment with the target dynamics. Combining the techniques above gives birth to our Selective Transition Correction (STC) algorithm, with its overall framework depicted in Figure 1. Empirical results on datasets with varied dynamics shifts show that STC exhibits competitive or better performance than prior strong baselines.

## 2 Related Work

**Offline Reinforcement Learning (RL).** Offline RL (Levine et al., 2020; Lange et al., 2012) addresses the problem of learning policies from fixed datasets without further environment interaction. A key challenge of offline RL lies in the extrapolation error (Fujimoto et al., 2019; Kumar et al., 2019). Offline RL methods

generally involve model-free methods (Xu et al., 2022; Wu et al., 2021; An et al., 2021; Lyu et al., 2022; Kostrikov et al., 2022; Yang et al., 2024; Tarasov et al., 2024; Yeom et al., 2024), and model-based methods (Yu et al., 2021; Matsushima et al., 2021; Yu et al., 2020; Kidambi et al., 2020; Qiao et al., 2024; Liu et al., 2025b). These approaches typically assume access to large-scale datasets from a single domain that closely matches the target environment. In contrast, we consider a more challenging setting where the target domain data is limited, and we aim to leverage supplementary source domain data to enhance policy performance.

**Domain Adaptation in RL.** We focus on the cross-domain policy adaptation problem in RL (Eysenbach et al., 2021; Xu et al., 2023; Lyu et al., 2024b), where the source domain and the target domain share the same state and action spaces but differ in their underlying dynamics. Effectively identifying and bridging the dynamics mismatch is a central challenge. Prior works have explored various techniques, including system identification (Clavera et al., 2018; Du et al., 2021; Xie et al., 2022), meta-RL (Nagabandi et al., 2018; Raileanu et al., 2020), domain randomization (Slaoui et al., 2019; Mehta et al., 2019; Vuong et al., 2019; Jiang et al., 2023), and imitation learning (Kim et al., 2019; Hejna et al., 2020; Fickinger et al., 2022), etc. However, the reliance on training environment distributions or expert demonstrations limits their practicality in many scenarios. Recent methods in online domain adaptation setting address this by learning dynamics models for both domains (Desai et al., 2020), value-guided data filtering (Xu et al., 2023) or dynamics-aware reward modification (Lyu et al., 2024a; Eysenbach et al., 2021; Van et al., 2024). In the context of cross-domain offline RL, existing approaches often involve reward penalization (Liu et al., 2022), dataset constraint (Liu et al., 2024), source transition filtering using mutual information (Wen et al., 2024) or optimal transport (Lyu et al., 2025), trajectory editing (Niu et al., 2024), data augmentation (Guo et al., 2025), flow matching (Kong et al., 2025), utilizing skill expansion and composition (Liu et al., 2025a), generating samples with a diffusion model (Van et al., 2025). These methods often focus less on source domain data that is not close to the target domain. In contrast, we propose to selectively correct source transitions into target domain transitions to better align with target domain dynamics.

## 3 Preliminaries

RL problems can be formulated as a Markov Decision Process (MDP), defined by $\mathcal{M} = (\mathcal{S}, \mathcal{A}, P, r, \gamma)$, where $\mathcal{S}, \mathcal{A}$ denote the state and action spaces, $P(s'|s,a)$ is the transition dynamics, $r(s,a) : \mathcal{S} \times \mathcal{A} \to \mathbb{R}$ is the scalar reward signal, $\gamma \in [0,1)$ is the discount factor. The objective of an RL agent is to learn a policy $\pi$ to maximize the expected discounted cumulative return $J(\pi) = \mathbb{E}_\pi \left[ \sum_{t=0}^\infty \gamma^t r(s_t, a_t) \right]$. Q-value is defined as $Q(s,a) := \mathbb{E}_\pi \left[ \sum_{t=0}^\infty \gamma^t r(s_t, a_t) | s, a \right]$. In cross-domain RL (Lyu et al., 2024b), we have a source domain $\mathcal{M}_{\mathrm{src}} = (\mathcal{S}, \mathcal{A}, P_{\mathrm{src}}, r, \gamma)$ and a target domain $\mathcal{M}_{\mathrm{tar}} = (\mathcal{S}, \mathcal{A}, P_{\mathrm{tar}}, r, \gamma)$, which share the state space, action space and reward function but differ in transition dynamics. We denote the transition dynamics in a domain $\mathcal{M}$ as $P_\mathcal{M}$, and $P_{\mathcal{M},t}^\pi(s)$ is the probability that the policy $\pi$ encounters the state $s$ at timestep $t$ in $\mathcal{M}$. Then we calculate the normalized probability that $\pi$ encounters the state-action pair $(s,a)$ in $\mathcal{M}$ as $\rho_\mathcal{M}^\pi(s,a) := (1-\gamma) \sum_{t=0}^\infty \gamma^t P_{\mathcal{M},t}^\pi(s)\pi(a|s)$. $P^\pi(\cdot|s) = \sum_a P(\cdot|s,a)\pi(a|s)$ is the transition dynamics induced by the policy $\pi$. We consider the offline setting where we have access to a static source domain dataset $D_{\mathrm{src}} = \{(s_{\mathrm{src}}^i, a_{\mathrm{src}}^i, r_{\mathrm{src}}^i, s_{\mathrm{src}}^{i+1})\}_{i=1}^N$ and a limited target domain dataset $D_{\mathrm{tar}} = \left\{ (s_{\mathrm{tar}}^i, a_{\mathrm{tar}}^i, r_{\mathrm{tar}}^i, s_{\mathrm{tar}}^{i+1}) \right\}_{i=1}^{N'}$, where $N$ and $N'$ are the dataset sizes. Cross-domain offline RL aims to leverage the mixed dataset $D_{\mathrm{src}} \bigcup D_{\mathrm{tar}}$ to acquire good performance in the target domain. We assume that each dataset corresponds to an empirical MDP, where the source domain dataset induces $\widehat{\mathcal{M}}_{\mathrm{src}}$ with dynamics $\widehat{P}_{\mathrm{src}}$, and the target domain dataset induces $\widehat{\mathcal{M}}_{\mathrm{tar}}$ with dynamics $\widehat{P}_{\mathrm{tar}}$. We denote behavior policies in the source and target domain datasets as $\mu_{\mathrm{src}}$ and $\mu_{\mathrm{tar}}$, the true transition dynamics in the source and target domain as $P_{\mathrm{src}}$ and $P_{\mathrm{tar}}$, and the transition dynamics in the corrected source domain dataset as $\widetilde{P}_{\mathrm{src}}$.

## 4 Selective Source Transition Correction

In this section, we describe key components in STC, which mainly contains two parts, (a) training an inverse policy model and a reward model in the target domain for source transition correction; (b) selectively correcting source domain transitions by using a forward dynamics model of the target domain dataset. We

theoretically analyze the dynamics discrepancy behavior and the value discrepancy behavior given corrected data. We also provide performance bounds of a policy on the corrected data and the true target domain.

## 4.1 Source Transition Correction

To align source domain transitions with the target dynamics, we introduce an inverse policy model and a reward model to correct actions and rewards of the source data to make them *target domain data*. The inverse policy model $f_{\text{inv}} : (s, s') \to a$ is optimized to predict the action that most likely incurs the observed next state under the target dynamics:

$$\mathcal{L}_{\text{inv}} = \mathbb{E}_{(s_{\text{tar}}, a_{\text{tar}}, s'_{\text{tar}}) \sim D_{\text{tar}}} \left[ \| f_{\text{inv}}(s_{\text{tar}}, s'_{\text{tar}}) - a_{\text{tar}} \|_2^2 \right] \tag{1}$$

As the inverse policy model captures the underlying dynamics of the target domain, we employ it as a surrogate to infer more target-consistent actions for state transitions from the source domain. Given a source transition $(s_{\text{src}}, a_{\text{src}}, s'_{\text{src}}) \in D_{\text{src}}$, where $a_{\text{src}}$ is the original source domain action, we apply the trained inverse model to produce a corrected action:

$$\hat{a}_{\text{src}} = f_{\text{inv}}(s_{\text{src}}, s'_{\text{src}}). \tag{2}$$

Since the reward of a transition is determined by both the state and the action, modifying the action necessitates a corresponding adjustment to the reward. To support this, we train a parametric reward model $r(s, a)$ to approximate the true reward function in the target domain using the available offline target dataset, which is optimized via:

$$\mathcal{L}_{\text{rew}} = \mathbb{E}_{(s_{\text{tar}}, a_{\text{tar}}, r_{\text{tar}}) \sim D_{\text{tar}}} \left[ (r(s_{\text{tar}}, a_{\text{tar}}) - r_{\text{tar}})^2 \right]. \tag{3}$$

The trained reward model is used to estimate the corrected reward for transitions in the source domain whose actions have been modified. For a transition $(s_{\text{src}}, a_{\text{src}}, s'_{\text{src}}, r_{\text{src}})$ and its corrected action $\hat{a}_{\text{src}}$, we apply a first-order Taylor expansion around the original action to approximate the corrected reward $\hat{r}_{\text{src}}$ as:

$$\hat{r}_{\text{src}} = r_{\text{src}} + \alpha \cdot \nabla_a r(s_{\text{src}}, a)^\top |_{a = a_{\text{src}}} (\hat{a}_{\text{src}} - a_{\text{src}}), \tag{4}$$

where $\nabla_a r(s_{\text{src}}, a)$ denotes the gradient of the reward model with respect to the action, and $\alpha$ is a tunable hyperparameter that scales the extent of reward adjustment. To ensure reward adjustment stability, the gradient is $\ell_2$-normalized and clipped within a bounded range. This correction leverages the local smoothness of the reward function in action space and enables efficient reward estimation without directly evaluating out-of-distribution (OOD) actions.

We then construct the candidate corrected source domain transition $(s_{\text{src}}, \hat{a}_{\text{src}}, s'_{\text{src}}, \hat{r}_{\text{src}})$. If the inverse policy model is sufficiently accurate, the corrected transition is expected to better align with the underlying dynamics of the target domain compared to the original transition $(s_{\text{src}}, a_{\text{src}}, s'_{\text{src}}, r_{\text{src}})$. This correction process allows for the effective reutilization of source domain data that would otherwise be incompatible, by substituting their actions with ones that are more consistent with the target domain dynamics.

## 4.2 Theoretical Analysis

To demonstrate the rationality of source transition correction, we provide a theoretical analysis given corrected source data. The following analysis is conducted under a tabular (finite state-action) MDP setting, and is intended to provide qualitative insight rather than a tight guarantee for the continuous control setting. It serves to provide intuition and partial justification for the proposed design under idealized conditions, and to clarify the role of transition correction in improving cross-domain alignment, thereby informing the design of the empirical correction mechanism. We first impose the following assumptions that are required for further theoretical analysis. These assumptions are common and widely used in RL. Due to space constraints, all proofs are deferred to Appendix A.

**Assumption 4.1.** There exists $\epsilon > 0$ such that $\| P_{\text{src}}(\cdot | s, a) - P_{\text{tar}}(\cdot | s, a) \| \leq \epsilon, \forall (s, a)$.

**Assumption 4.2.** The estimated inverse policy model $\hat{\pi}_{\text{inv}}$ well approximates the true empirical inverse policy model $\pi_{\text{inv}}^{\text{tar}}$ such that the error between the empirical forward policy models in the corrected data and the target domain dataset is bounded, i.e., $\mathbb{E}\left[ \| \hat{\pi}(s) - \mu_{\text{tar}}(s) \|_1 \right] \leq \kappa$.

**Assumption 4.3.** The reward function is bounded and is $L_r$-smooth, i.e., $\forall (s, a), \|\nabla_a r(s, a)\| \leq L_r, |r(s, a)| \leq r_{\max}$.

Assumption 4.1 requires that the dynamics discrepancy between the source domain and the target domain is bounded, and Assumption 4.2 assumes that the estimated inverse policy model well-fit the target domain. Theorem 4.4 depicts the dynamics discrepancy between the corrected source domain dataset and the target domain dataset.

**Theorem 4.4.** *Denote the corrected source domain transition dynamics as $\widetilde{P}_{\mathrm{src}}$, then under Assumption 4.1 and 4.2, the deviation between the corrected dynamics and the empirical target domain dynamics $\widehat{P}_{\mathrm{tar}}(\cdot|s, a)$ is bounded:*

$$\|\widetilde{P}_{\mathrm{src}}(\cdot|s, a) - \widehat{P}_{\mathrm{tar}}(\cdot|s, a)\| \leq \kappa + \epsilon.$$

Furthermore, we show the value discrepancy of $Q^\pi(s, a)$ given a policy $\pi$ between the corrected data and raw data.

**Theorem 4.5.** *Given Assumption 4.3 and assume that the source domain rewards are corrected via $\hat{r}(s_{\mathrm{src}}, a_{\mathrm{src}}) = r(s_{\mathrm{src}}, a_{\mathrm{src}}) + \nabla_a r(s_{\mathrm{src}}, a)^\top|_{a=a_{\mathrm{src}}}(\hat{a}_{\mathrm{src}} - a_{\mathrm{src}})$, where $\hat{a}_{\mathrm{src}} \sim \mu_{\mathrm{tar}}(\cdot|s_{\mathrm{src}})$. Then given any $(s, a)$, the deviation of Q-values on the corrected empirical source domain $\widetilde{\mathcal{M}}_{\mathrm{src}}$ and the raw empirical source domain $\widehat{\mathcal{M}}_{\mathrm{src}}$ is bounded:*

$$\left| Q^\pi_{\widetilde{\mathcal{M}}_{\mathrm{src}}}(s, a) - Q^\pi_{\widehat{\mathcal{M}}_{\mathrm{src}}}(s, a) \right| \leq \frac{2L_r}{1 - \gamma} D_{\mathrm{TV}}(\mu_{\mathrm{src}} \| \mu_{\mathrm{tar}}).$$

Theorem 4.5 shows that the deviation of Q-values on $\widehat{\mathcal{M}}_{\mathrm{src}}$ and $\widetilde{\mathcal{M}}_{\mathrm{src}}$ is bounded by the total variation deviation between the behavior policies in the source domain dataset and target domain dataset. This ensures that the corrected value function well reflects the dynamics discrepancy between the two domains and verifies the necessity of reward correction. To see how transition correction affects the performance of the agent, we derive a concrete performance bound of a policy given the corrected source domain data and the true target domain in Theorem 4.6.

**Theorem 4.6** (Finite data bound)**.** *Denote $\widetilde{\mathcal{M}}_{\mathrm{src}}$ as the corrected empirical source domain MDP, $n$ is the size of the target domain dataset, $C_1 = \frac{\gamma r_{\max}|\mathcal{S}|}{\sqrt{2}(1-\gamma)^2}$, $C_2 = |\mathcal{S} \times \mathcal{A} \times \mathcal{S}|$. Then under Assumption 4.1-4.3, for any policy $\pi$, the following bound holds with probability at least $1 - \delta$:*

$$J_{\widetilde{\mathcal{M}}_{\mathrm{src}}}(\pi) - J_{\mathcal{M}_{\mathrm{tar}}}(\pi) \geq -\frac{\gamma r_{\max}(\kappa + \epsilon)}{(1 - \gamma)^2} - C_1\sqrt{\frac{1}{n}\ln\frac{2C_2}{\delta}}.$$

The above bound indicates that the performance difference of a policy $\pi$ in two domains is decided by the policy estimation error $\kappa$ (other components are constants). The lower bound becomes tight (i.e, the policy can closely match its performance in the true target domain by using corrected source domain data) if the inverse policy is well-trained (i.e., $\kappa$ is small) and large vice versa. It also highlights the necessity of training a good inverse policy model.

### 4.3 Selective Correction Mechanism

Since $D_{\mathrm{tar}}$ contains limited data, the learned inverse policy model may be less reliable in OOD regions. This hypothesis is supported by preliminary experiments, where we attempt to apply action correction uniformly across all source domain transitions. While this approach yields performance improvements in some environments, it leads to significant degradation in others. This suggests that, in certain cases, corrections produced by the inverse policy and reward model may not provide consistent benefits and could potentially introduce errors for policy learning.

To address this challenge, we introduce a selective correction mechanism that operates in a conservative manner, correcting source-domain samples only when the model is (comparatively) confident about its prediction. To fulfill that, we quantify the dynamic discrepancy between a transition and the target domain and use it as a metric to decide whether the inverse policy model can be reliable. It motivates us to additionally

train a forward dynamics model upon the target domain dataset, which predicts the next state difference given the current state-action pair, by minimizing the following loss:

$$\mathcal{L}_{\text{fwd}} = \mathbb{E}_{(s_{\text{tar}}, a_{\text{tar}}, s'_{\text{tar}}) \sim D_{\text{tar}}} \left[ \| f_{\text{fwd}}(s_{\text{tar}}, a_{\text{tar}}) - (s'_{\text{tar}} - s_{\text{tar}}) \|_2^2 \right] \tag{5}$$

For the original source transition $(s_{\text{src}}, a_{\text{src}}, s'_{\text{src}}, r_{\text{src}})$ and its corrected counterpart $(s_{\text{src}}, \hat{a}_{\text{src}}, s'_{\text{src}}, \hat{r}_{\text{src}})$, we compute their respective *dynamics discrepancies* with respect to the target domain, denoted by $\varepsilon_{\text{orig}}$ and $\varepsilon_{\text{corr}}$. These discrepancies are defined as the prediction errors of a forward dynamics model trained on the target dataset:

$$\varepsilon_{\text{orig}} = \| f_{\text{fwd}}(s_{\text{src}}, a_{\text{src}}) - (s'_{\text{src}} - s_{\text{src}}) \|^2,$$
$$\varepsilon_{\text{corr}} = \| f_{\text{fwd}}(s_{\text{src}}, \hat{a}_{\text{src}}) - (s'_{\text{src}} - s_{\text{src}}) \|^2. \tag{6}$$

The corrected transition is adopted only if its dynamics discrepancy is substantially smaller than that of the original transition, formally defined as:

$$\tilde{\tau}_{\text{src}} = \begin{cases} (s_{\text{src}}, \hat{a}_{\text{src}}, s'_{\text{src}}, \hat{r}_{\text{src}}), & \text{if } \varepsilon_{\text{corr}} < \lambda \cdot \varepsilon_{\text{orig}}, \\ (s_{\text{src}}, a_{\text{src}}, s'_{\text{src}}, r_{\text{src}}), & \text{otherwise,} \end{cases} \tag{7}$$

where $\lambda$ is a tunable threshold hyperparameter. Then we construct the corrected source domain dataset $\widetilde{D}_{\text{src}}$ using such selectively corrected transitions:

$$\widetilde{D}_{\text{src}} = \{ \tilde{\tau}_{\text{src}} \mid (s_{\text{src}}, a_{\text{src}}, r_{\text{src}}, s'_{\text{src}}) \in D_{src} \}. \tag{8}$$

Finally, we use $\widetilde{D}_{\text{src}}$ together with the original target data $D_{\text{tar}}$ to train the final policy, $D_{\text{train}} = D_{\text{tar}} \bigcup \widetilde{D}_{\text{src}}$. Note that when the correction is rejected, the original source transition is retained for training since we believe there are still some underlying shared behavior embedded in those data that can be beneficial for policy learning. By mutually constraining each other, the inverse and forward models help prevent performance degradation caused by errors from a single model. This selective correction mechanism enables more stable and conservative corrections, producing more reliable data for offline learning.

We remark that although the target-domain dataset is limited, both the inverse policy model and the forward dynamics model can be learned in a relatively reliable manner. In our implementation, the models are pretrained for 50k steps before policy learning. Their prediction losses consistently decrease during training and converge to small magnitudes, with the inverse model reaching on the order of $10^{-4}$ and the forward model stabilizing around $10^{-2}$, indicating reliable data fitting at this scale. This further supports the validity of STC under limited target dataset setting.

### 4.4 Actor-Critic Learning

After constructing the training dataset $D_{\text{train}}$, we train the policy under an offline actor-critic framework. We optimize the Q-function $Q_\theta$ via temporal difference (TD) learning:

$$\mathcal{L}_Q(\theta) = \mathbb{E}_{(s, a, r, s') \sim D_{\text{train}}} \left[ (Q_\theta(s, a) - y)^2 \right], \tag{9}$$

where $y = r + \gamma \min_{j=1,2} Q_{\theta_j^-}(s', \pi(s'))$ is the target value, $\theta^-$ is the target $Q$-network parameters. To mitigate the distribution shift issue and prevent the agent from exploiting OOD actions, we regularize the policy learning with the Q-value-weighted behavior cloning:

$$\mathcal{L}_\pi(\phi) = -\mathbb{E}_{(s, a) \sim D_{\text{train}}} \left[ \eta Q_\theta(s, \pi_\phi(s)) - \beta \cdot \exp\left( \eta Q_\theta(s, a) \right) \| \pi_\phi(s) - a \|_2^2 \right], \tag{10}$$

where $\eta = \frac{1}{\frac{1}{n} \sum_{i=1}^{n} |Q_\theta(s_i, a_i)|}$ is a batch-wise scaling factor and $n$ denotes the batch size, $\beta$ is a hyperparameter used to balance the behavior regularization error and Q-loss. We use Q-values as weights for behavior cloning loss to inform the agent the importance of each transition, akin to IQL (Kostrikov et al., 2022). We summarize the pseudocode for STC in Algorithm 1, which can be found in Appendix D.

# 5 Experiments

In this section, we evaluate the effectiveness of our method for offline policy adaptation through experiments in varied environments with dynamics shifts and dataset qualities. We additionally conduct a visualization study to validate the reliability of STC in correcting source transitions. Moreover, ablation studies on key hyperparameters are performed to further understand the hyperparameter sensitivity of STC. Considering the page limit, please check more experimental results in Appendix E.

## 5.1 Main Results

**Tasks and datasets.** We consider three kinds of dynamics shifts, including gravity shift, friction shift, and morphology shift, for four tasks (*ant, hopper, halfcheetah, walker2d*) from ODRL (Lyu et al., 2025) to comprehensively evaluate the cross-domain offline policy adaptation ability. The gravity shift modifies the strength of the gravity while keeping its direction unchanged. The friction shift is introduced by adjusting the static, dynamic, and rolling friction components. The morphology shift modifies the size of specific limbs or torsos of the simulated robot in the target domain. As we focus on cross-domain offline RL setting with limited target domain data, we use the original environments as source domain and use those modified environments as target domain. We adopt the MuJoCo "-v2" datasets from D4RL (Fu et al., 2020) for source domain datasets and use ODRL datasets in modified environments as target domain datasets (only **5000** transitions). We adopt ODRL medium and expert datasets and construct medium-expert datasets by selecting 2 trajectories from medium datasets and 3 trajectories from expert datasets. For source domain datasets, we adopt medium, medium-replay and medium-expert datasets. We conduct experiments across various combinations of data qualities and dynamics shifts. All algorithms are trained for 1M gradient steps across 5 random seeds.

**Baselines.** We consider the following typical baselines: **IQL** (Kostrikov et al., 2022) that is trained on the combined source and target dataset; **DARA** (Liu et al., 2022) that trains domain classifiers to impose penalties on source domain rewards; **BOSA** (Liu et al., 2024) that addresses the OOD issue through support-constrained policy and value optimization; **SRPO** (Xue et al., 2024) that modifies rewards based on the stationary state distribution; **IGDF** (Wen et al., 2024) that introduces a contrastive score function to selectively share source transitions; **OTDF** (Lyu et al., 2025) that filters source data via optimal transport.

**Metrics.** To ensure that the results are interpretable across different tasks, we follow ODRL (Lyu et al., 2024b) and adopt the normalized score (NS) in the target domain as the evaluation metric, defined as $\text{NS} = \frac{J - J_r}{J_e - J_r} \times 100$, where $J$, $J_e$, and $J_r$ denote the returns of the learned, expert, and random policies in the target domain, respectively.

**Results.** We summarize the performance of all methods under gravity shift and morphology shift in Table 1. Due to space limitations, results for friction shift setting are deferred to Appendix E.1. For each task, we vary the quality of the source domain data to evaluate the robustness of different methods under varied dynamics shifts and data quality combinations. Our results show that STC consistently outperforms all baselines in most tasks and achieves a total normalized score **1045.2**. In particular, compared to IQL, which directly learns from the mixed dataset without any adaptation, STC achieves a notable improvement of 54%, highlighting the effectiveness of selectively correcting source domain transitions. STC achieves the highest normalized score on **18 out of all 24 tasks**. OTDF is the best-performing baseline excluding STC, and STC beats OTDF in overall average performance by 27%. While STC slightly underperforms other top-performing methods on several tasks, such as some in *walker2d-gravity-shift* environment, the gap is marginal and does not indicate a significant weakness. When the domain shift is relatively mild or the source transitions are already well aligned with the target dynamics, the additional benefit of correction can be limited. More generally, STC tends to be more effective when correction improves cross-domain alignment without introducing excessive perturbation.

We find that some policy adaptation methods offer limited improvement over IQL (as also observed in (Lyu et al., 2025)), likely due to the limited quantity (5000) of available target transitions, which increases the difficulty of effective adaptation. Methods like DARA and SRPO may fail to learn reliable domain classifiers

Table 1: **Performance comparison under distinct dynamics shift**. half = halfcheetah, hopp = hopper, walk = walker2d, med = medium, r = replay, e = expert. **Source** means the source domain dataset, and **Target** indicates the target domain dataset quality. The normalized average scores in the target domain across 5 seeds are reported and $\pm$ captures the standard deviation. We highlight the best cell.

| Type | Source | Target | IQL | DARA | BOSA | SRPO | IGDF | OTDF | STC (ours) |
|---|---|---|---|---|---|---|---|---|---|
| Gravity | half-med | med | 39.6±3.3 | 41.2±3.9 | 38.9±4.0 | 36.9±4.5 | 36.6±5.5 | 40.7±7.7 | **42.4**±5.3 |
| | half-med-r | med | 20.1±5.0 | 17.6±6.2 | 20.0±4.9 | 17.5±5.2 | 14.4±2.2 | 21.5±6.5 | **26.7**±2.2 |
| | half-med-e | med | 38.6±6.0 | 37.8±3.3 | 41.8±5.1 | 42.5±2.3 | 37.7±7.3 | 39.5±3.5 | 39.2±4.2 |
| | hopp-med | med | 11.2±1.1 | 17.3±3.8 | 15.2±3.3 | 12.4±1.0 | 15.3±3.5 | 32.4±8.0 | **43.4** ±6.1 |
| | hopp-med-r | med | 13.9±2.9 | 10.7±4.3 | 3.3±1.9 | 14.0±2.6 | 15.3±4.4 | 31.1±13.4 | **36.8**±17.8 |
| | hopp-med-e | med | 19.1±6.6 | 18.5±12.3 | 15.9±5.9 | 19.7±8.5 | 22.3±5.4 | 26.4±10.1 | **45.3**±7.5 |
| | walk-med | med | 28.1±12.9 | 28.4±13.7 | 38.0±11.2 | 21.4±7.0 | 22.1±8.4 | 36.6±2.3 | **41.6**±4.0 |
| | walk-med-r | med | 14.6±2.5 | 14.1±6.1 | 7.6±5.8 | 17.9±3.8 | 11.6±4.6 | **32.7**±7.0 | 29.0 ±1.9 |
| | walk-med-e | med | 39.9±13.1 | 41.6±13.0 | 32.3±7.2 | 46.4±3.5 | 33.8±3.1 | 30.2±9.8 | 34.9 ±9.1 |
| | ant-med | med | 10.2±1.8 | 9.4±0.9 | 12.4±2.0 | 11.7±1.0 | 11.3±1.3 | **45.1**±12.4 | 42.6±8.4 |
| | ant-med-r | med | 18.9±2.6 | 21.7±2.1 | 13.9±1.5 | 18.7±1.7 | 19.6±1.0 | 29.6±10.7 | **40.9**±5.6 |
| | ant-med-e | med | 9.8±2.4 | 8.1±1.8 | 8.1±3.0 | 8.4±2.1 | 8.9±1.5 | 18.6±11.9 | **39.2**±9.2 |
| Morph | half-med | med | **24.5**±2.4 | 21.0±3.9 | 24.2±5.6 | 18.1±1.8 | 23.7±3.4 | 21.1±7.6 | 19.5±2.2 |
| | half-med-r | med | 11.0±1.2 | 9.5±2.3 | 4.7±2.9 | 8.9±1.2 | 9.2±0.6 | 6.5±1.4 | **13.0**±5.3 |
| | half-med-e | med | 21.1±2.8 | 19.2±2.2 | **23.2**±3.9 | 21.1±1.9 | 18.6±1.3 | 20.8±2.5 | 16.8±8.8 |
| | hopp-med | med | 15.9±6.8 | 17.8±10.1 | 12.8±0.1 | 21.7±7.7 | 25.3±9.7 | 16.4±7.1 | **43.1**±23.9 |
| | hopp-med-r | med | 12.9±0.3 | 12.8±0.1 | 2.0±1.2 | 12.4±0.7 | 12.5±1.7 | 13.3±0.1 | **22.9**±6.1 |
| | hopp-med-e | med | 14.9±3.1 | 11.1±5.6 | 14.4±1.8 | 16.6±1.9 | 18.3±7.5 | 25.4 ±9.4 | **53.4**±20.8 |
| | walk-med | med | 31.5±8.6 | 35.0±10.8 | 26.7±6.6 | 38.6±5.1 | 38.5±8.4 | 42.5±3.1 | **56.7**±8.1 |
| | walk-med-r | med | 41.5±3.0 | 38.5±7.9 | 15.3±7.6 | 36.0±4.4 | 24.2±8.6 | 17.9±13.4 | **63.1**±8.0 |
| | walk-med-e | med | 32.8±4.3 | 41.4±10.9 | 45.1±13.7 | 39.8±14.3 | 37.9±4.2 | 55.3±2.2 | **62.1**±8.1 |
| | ant-med | med | 71.4±2.4 | 71.5±6.8 | 54.8±13.2 | 72.8±2.2 | 71.8±2.7 | 75.1±2.3 | **77.2**±2.8 |
| | ant-med-r | med | 65.9±5.5 | 62.3±8.2 | 15.2±2.3 | 59.3±4.0 | 65.0±5.3 | 63.1±5.9 | **76.2**±2.6 |
| | ant-med-e | med | 70.2±7.3 | 64.3±5.8 | 64.0±6.0 | 68.5±4.4 | 66.8±11.0 | 76.4±1.9 | **79.3**±0.2 |
| | Total Score | | 677.6 | 670.9 | 549.8 | 681.1 | 660.8 | 818.1 | **1045.2** |

under such conditions, resulting in inaccurate reward modification. OTDF achieves clear improvements over IQL by using optimal transport, but it still falls short compared to STC. It indicates that selective source transition correction can be more powerful than source data filtering, which allows for more effective reuse of potentially valuable transitions. Meanwhile, the bidirectional dynamics-based selection mechanism helps mitigate the negative effects of potential model training bias. Moreover, the training efficiency of STC remains competitive with that of other baselines (see Appendix 6 for details). Considering the performance gains achieved, we deem the extra training effort of the auxiliary models acceptable.

## 5.2 Influence of Target Domain Dataset Quality

To evaluate the robustness of our method under varying target data quality, we conduct experiments using target domain datasets of different qualities. As demonstrated in Table 2, STC exhibits superior performance across varying data quality settings, sometimes outperforming baseline methods by a large margin. In general, STC achieves the highest score on 12 out of the 20 tasks and a total score of **820.8**, which beats all baseline methods and surpasses the second best baseline OTDF by **13.5%**. These empirical results clearly show the strong adaptability and effectiveness of STC under diverse target domain dataset qualities.

## 5.3 Influence of Target Domain Dataset Size

Our method has demonstrated strong effectiveness, even when only a limited amount of target domain data is available. To further validate the general applicability of STC, we systematically vary the size of the target domain dataset and evaluate its impact on performance. Specifically, we train all methods using different amounts of target transitions (e.g., 5k, 100k) across several environments and report the normalized scores in Table 3. We observe that the performance of all methods generally improves as the dataset size increases. However, STC consistently outperforms the baselines across most tasks, regardless of whether the

Table 2: **Performance comparison under distinct target dataset qualities**. half = halfcheetah, hopp = hopper, walk = walker2d, med = medium, e = expert. **Source** means the source domain dataset, and **Target** indicates the target domain dataset quality. The normalized average scores in the target domain across 5 seeds are reported and ± captures the standard deviation. We highlight the best cell.

| Type | Source | Target | IQL | DARA | BOSA | SRPO | IGDF | OTDF | STC (ours) |
|---|---|---|---|---|---|---|---|---|---|
| Gravity | half-med | med | 39.6±3.3 | 41.2±3.9 | 38.9±4.0 | 36.9±4.5 | 36.6±5.5 | 40.7±7.7 | **42.4±5.3** |
| | half-med | med-e | 39.6±3.7 | **40.7±2.8** | 40.4±3.0 | **40.7±2.3** | 38.7±6.2 | 28.6±3.2 | 37.3±1.5 |
| | half-med | expert | 42.4±3.8 | 39.8±4.4 | 40.5±3.9 | 39.4±1.6 | 39.6±4.6 | 36.1±5.3 | **43.1±5.7** |
| | hopp-med | med | 11.2±1.1 | 17.3±3.8 | 15.2±3.3 | 12.4±1.0 | 15.3±3.5 | 32.4±8.0 | **43.4 ±6.1** |
| | hopp-med | med-e | 14.7±3.6 | 15.4±2.5 | 21.1±9.3 | 14.2±1.8 | 15.1±3.6 | **24.2±3.6** | 23.4±6.4 |
| | hopp-med | expert | 12.5±1.6 | 19.3±10.5 | 12.7±1.7 | 11.8±0.9 | 14.8±4.0 | 33.7±7.8 | **37.1±14.1** |
| | walk-med | med | 28.1±12.9 | 28.4±13.7 | 38.0±11.2 | 21.4±7.0 | 22.1±8.4 | 36.6±2.3 | **41.6±4.0** |
| | walk-med | med-e | 35.7±4.7 | 30.7±9.7 | 40.9±7.2 | 34.0±9.9 | 35.4±9.1 | **44.8±7.5** | 36.9±6.5 |
| | walk-med | expert | 37.3±8.0 | 36.0±7.0 | 41.3±8.6 | 39.5±3.8 | 36.2±13.6 | 44.0±4.0 | **49.9±10.1** |
| | ant-med | med | 10.2±1.8 | 9.4±0.9 | 12.4±2.0 | 11.7±1.0 | 11.3±1.3 | **45.1±12.4** | 42.6±8.4 |
| | ant-med | med-e | 9.4±1.2 | 10.0±0.9 | 11.6±1.3 | 10.2±1.2 | 9.4±1.4 | **33.9±5.4** | 23.2±6.1 |
| | ant-med | expert | 10.2±0.3 | 9.8±0.6 | 11.8±0.4 | 9.5±0.6 | 9.7±1.6 | 33.2±9.0 | **35.6±5.6** |
| Morph | half-med | med | **24.5±2.4** | 21.0±3.9 | 24.2±5.6 | 18.1±1.8 | 23.7±3.4 | 21.1±7.6 | 19.5±2.2 |
| | half-med | expert | 11.1±1.5 | **11.3±0.6** | 9.1±1.7 | 10.1±1.6 | 10.1±0.8 | 7.4±2.3 | 7.7±1.2 |
| | hopp-med | med | 15.9±6.8 | 17.8±10.1 | 12.8±0.1 | 21.7±7.7 | 25.3±9.7 | 16.4±7.1 | **43.1±23.9** |
| | hopp-med | expert | 15.5±4.1 | 13.0±1.4 | 4.5±4.0 | 10.4±3.1 | 13.4±2.0 | 14.0 ±4.0 | **53.2±50.3** |
| | walk-med | med | 31.5±8.6 | 35.0±10.8 | 26.7±6.6 | 38.6±5.1 | 38.5±8.4 | 42.5±3.1 | **56.7±8.1** |
| | walk-med | expert | 37.0±6.0 | 43.7±5.5 | 31.3±8.6 | 43.8±8.4 | 38.5±4.3 | **49.9±5.4** | 32.9±7.3 |
| | ant-med | med | 71.4±2.4 | 71.5±6.8 | 54.8±13.2 | 72.8±2.2 | 71.8±2.7 | 75.1±2.3 | **77.2±2.8** |
| | ant-med | expert | 63.9±8.1 | 68.9±7.6 | 47.5±11.9 | 59.0±4.7 | 66.1±4.0 | 63.8±6.4 | **74.1±21.2** |
| | Total Score | | 561.6 | 580.2 | 535.8 | 556.1 | 571.7 | 723.3 | **820.8** |

Table 3: **Performance comparison under distinct target domain dataset size**. med = medium. The **Source** column means the source domain dataset, and the **Size** column indicates the size of target domain dataset. The normalized average scores in the target domain across 5 seeds are reported and ± captures the standard deviation. We highlight the best cell.

| Type | Source | Size | IQL | DARA | BOSA | SRPO | IGDF | OTDF | STC (ours) |
|---|---|---|---|---|---|---|---|---|---|
| Gravity | hopper-med | 5k | 11.2±1.1 | 17.3±3.8 | 15.2±3.3 | 12.4±1.0 | 15.3±3.5 | 32.4±8.0 | **43.4 ±6.1** |
| | hopper-med | 100k | 18.0±4.4 | 19.4±14.6 | 15.8±6.6 | 18.1±6.6 | 17.0±6.2 | 47.1±11.4 | **66.1±4.8** |
| | walker2d-med | 5k | 28.1±12.9 | 28.4±13.7 | 38.0±11.2 | 21.4±7.0 | 22.1±8.4 | 36.6±2.3 | **41.6±4.0** |
| | walker2d-med | 100k | 33.0±7.9 | 28.4±5.0 | 40.3±10.9 | 33.9±8.1 | 41.9±5.8 | 42.8±5.2 | **45.2±3.3** |
| Morph | hopper-med | 5k | 15.9±6.8 | 17.8±10.1 | 12.8±0.1 | 21.7±7.7 | 25.3±9.7 | 16.4±7.1 | **43.1±23.9** |
| | hopper-med | 100k | 21.5±10.7 | 18.6±8.5 | 12.8±0.1 | 26.6±9.9 | 31.3±13.9 | 30.5±18.3 | **57.8±22.5** |
| | walker2d-med | 5k | 31.5±8.6 | 35.0±10.8 | 26.7±6.6 | 38.6±5.1 | 38.5±8.4 | 42.5±3.1 | **56.7±8.1** |
| | walker2d-med | 100k | 75.6±6.6 | **79.1±3.8** | 44.8±11.8 | 69.6±5.1 | 75.6±5.8 | 64.2±4.5 | 67.4±6.1 |
| | Total Score | | 234.9 | 243.9 | 206.5 | 242.3 | 267.0 | 312.4 | **421.3** |

dataset size is limited (e.g., 5k) or relatively large (e.g., 100k). This demonstrates that our method remains effective under limited data and scales efficiently with larger datasets, highlighting STC's robustness and data efficiency in cross-domain adaptation.

## 5.4 STC Can Correct Source Samples Reliably

To evaluate the reliability of STC in correcting source transitions, we conduct a visualization study to assess whether the corrected source transition distribution better aligns with the target domain. Specifically, we take the *hopper* environment with the gravity shift and the *walker2d* environment with the morphology shift as illustrative examples, and additional visualizations on other environments are provided in Appendix E.2. We first apply STC to original source transitions to obtain the corrected source transitions. For each transition in the target dataset, we identify its nearest neighbor using a KD-Tree in the original source dataset based on the state transition pair $(s, s')$, and extract the corresponding original action $a_{\mathrm{src}}$. We then

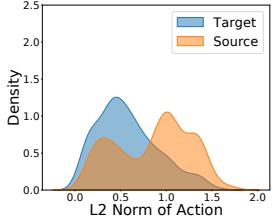 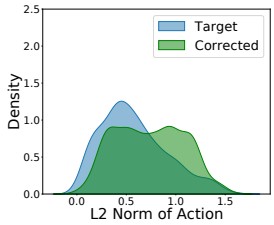

(a) Comparison in *hopper-gravity* environment.

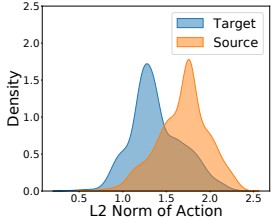 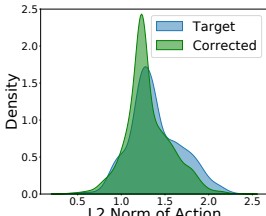

(b) Comparison in *walker2d-morph* environment.

Figure 2: **Action distribution comparison in (a) the hopper (gravity shift) and (b) the walker2d (morphology shift) environments.** In each subplot, the left panel shows KDE curves comparing original source domain actions and target domain actions, while the right panel shows KDE curves comparing STC-corrected source actions with target actions.

Table 4: **Ablation study on the effect of the selective mechanism for source transition correction. Source** means the source domain dataset, and **Target** indicates the target domain dataset quality. The normalized average scores in the target domain across 5 seeds are reported and ± captures the standard deviation.

| Type | Source | Target | w/o Selection | STC (ours) |
|---|---|---|---|---|
| Gravity | walker2d-medium | medium | 39.0±4.1 | **41.6±4.0** |
| | halfcheetah-medium | medium | 27.6±5.6 | **42.4±5.3** |
| | hopper-medium | medium | 37.9±5.5 | **43.4±6.1** |
| Morph | halfcheetah-medium | medium | 4.2±8.4 | **19.5±2.2** |
| | walker2d-medium | medium | **59.7±8.3** | 56.7±8.1 |
| | hopper-medium | medium | 42.8±7.7 | **43.1±23.9** |
| Total Score | | | 211.2 | **246.8** |

locate the transition with the same $(s, s')$ in the corrected source dataset and extract the corrected action $\hat{a}_{\text{src}}$. Finally, we plot the kernel density estimation (KDE) curves of both $a_{\text{src}}$ and $\hat{a}_{\text{src}}$ and compare them with the action distribution of the target domain dataset. As shown in both Figure 2a and 2b, the corrected source action distribution (the green curve) aligns more closely with the target domain distribution (the blue curve) compared to the original source action distribution (the orange curve), indicating that STC effectively aligns corrected source transitions with the target dynamics.

## 5.5 Component Analysis of STC

We conduct ablation studies to evaluate the contributions of the two components in STC: the selective mechanism and reward correction.

**Selective mechanism.** We first compare the full STC method with a variant that removes the forward-model-based selection mechanism. As shown in Table 4, removing this component leads to consistent performance degradation on most tasks, such as *halfcheetah-gravity, halfcheetah-morph*, and *hopper-gravity*. This indicates that directly applying inverse corrections can introduce mismatched or noisy action modifications, while the selective mechanism effectively filters out detrimental corrections and retains transitions that are more consistent with the target dynamics. On a few tasks, the performance gap is marginal, suggesting that inverse corrections are already sufficiently accurate in these environments, thereby limiting the benefit of additional filtering. Overall, the results demonstrate the importance of the selective mechanism in ensuring the reliability of transition correction under dynamics mismatch..

**Reward Correction.** We further assess the role of reward correction in STC via an ablation that removes this component while retaining action correction. The results are summarized in Table 5. We observe that action correction alone accounts for most of the performance improvement, while reward correction provides

Table 5: **Ablation study on the effect of reward correction. Action Only** denotes the variant that retains action correction while removing reward correction in STC.

| Type | Source | Target | Action Only | STC (ours) |
|------|--------|--------|-------------|------------|
| Gravity | walker2d-medium | medium | 35.9±2.8 | **41.6±4.0** |
| | halfcheetah-medium | medium | 38.1±1.0 | **42.4±5.3** |
| | hopper-medium | medium | 40.5±5.9 | **43.4±6.1** |
| Morph | walker2d-medium | medium | 56.4±9.9 | **56.7±8.1** |
| | halfcheetah-medium | medium | 17.4±5.9 | **19.5±2.2** |
| | hopper-medium | medium | 33.6±12.1 | **43.1±23.9** |
| Total Score | | | 221.9 | **246.8** |

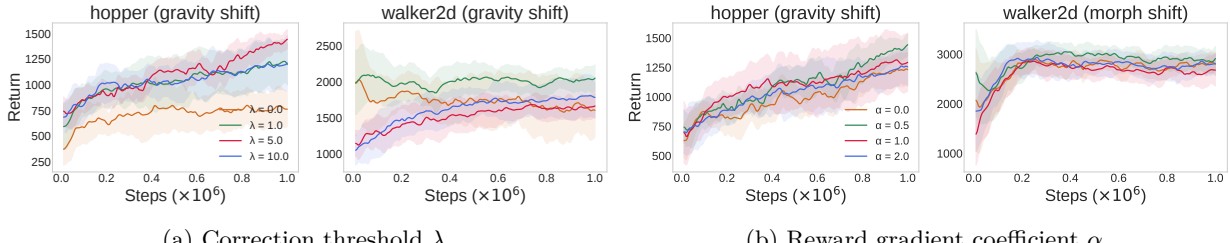

(a) Correction threshold $\lambda$.        (b) Reward gradient coefficient $\alpha$.

Figure 3: **Parameter study of STC**. We report target domain return results in two shift tasks. The shaded region captures the standard deviation.

an additional and generally consistent gain. These observations suggest that reward correction complements action correction by mitigating the bias introduced by modified transitions, thereby improving training stability and final performance.

Overall, both components contribute consistent improvements, leading to better performance and stability of STC under dynamics mismatch.

### 5.6 Parameter Study

We conduct an ablation study to investigate the sensitivity of STC to key hyperparameters: the correction threshold $\lambda$ and the reward gradient coefficient $\alpha$.

**Correction threshold $\lambda$.** The coefficient $\lambda$ decides whether the source domain transition should be corrected, i.e., a corrected transition is adopted only if $\varepsilon_{\text{corr}} < \lambda \cdot \varepsilon_{\text{orig}}$. Smaller $\lambda$ enforces stricter alignment with target dynamics but may limit the number of accepted corrected transitions. In contrast, larger $\lambda$ allows more corrections but can introduce misaligned samples. Therefore, selecting an appropriate $\lambda$ is crucial for balancing correction quality and data coverage. We run STC with $\lambda \in \{0, 1.0, 5.0, 10.0\}$, and the results in Figure 3a show that no correction ($\lambda = 0$) leads to unsatisfying performance, while different tasks achieve the best results with different $\lambda$. Across all tasks, we adopt $\lambda = 1.0$ or $5.0$ to achieve favorable performance.

**Reward gradient coefficient $\alpha$.** The coefficient $\alpha$ controls the extent of reward adjustment. A small $\alpha$ may lead to insufficient reward adjustment, failing to capture the influence of the action change; a large $\alpha$ may amplify model noise or cause over-adjustment. We conduct experiments with $\alpha \in \{0.0, 0.5, 1.0, 2.0\}$ and present results in Figure 3b. We observe that STC is less sensitive to $\alpha$ compared to $\lambda$. As setting $\alpha = 0.5$ yields the best performance across most tasks, we fix this value throughout all experiments.

## 6 Computational Overhead

In this section, we compare the computational overhead of different cross-domain offline RL algorithms. STC requires training the forward dynamics model and the inverse policy model, which introduces additional

computational cost. Many existing cross-domain offline RL methods share this issue. For example, DARA Liu et al. (2022) requires training domain classifiers and constructing reward penalties, while OTDF Lyu et al. (2025) needs to solve optimal transport problem. To quantify the computational overhead, we measure the runtime of STC and these baseline methods (DARA (Liu et al., 2022), OTDF (Lyu et al., 2025), SRPO (Xue et al., 2024)) across two representative tasks under identical compute infrastructure (Appendix G), all trained for 1M gradient steps. As summarized in Table 6, the total runtime of STC remains competitive with the baselines. Since our algorithm achieves substantially better performance than the baselines, we consider this computational overhead acceptable.

Table 6: Comparison of wall-clock training time (HH:MM:SS) for different methods.

| Task | DARA | OTDF | SRPO | STC (Total) |
|---|---|---|---|---|
| Hopper-gravity-shift | 1:59:04 | 2:37:23 | 1:46:05 | 1:49:39 |
| HalfCheetah-gravity-shift | 2:08:29 | 2:50:07 | 2:02:36 | 2:02:51 |

## 7 Conclusion

In this paper, we address the challenge of offline policy adaptation across domains with dynamics mismatch. Unlike existing methods that typically mitigate the mismatch through data filtering or reward penalties, we directly correct source domain transitions to better align with the target dynamics. Specifically, we propose Selective Transition Correction (STC), a framework that modifies source transitions by leveraging the inverse policy model and the forward dynamics model trained on the target domain. The inverse model generates corrected actions and rewards, while the forward model is used to select transitions that are more consistent with the target dynamics. Extensive experiments on benchmarks with varying data qualities and types of dynamics shift demonstrate that STC consistently outperforms existing baselines, often with substantial performance gains. Our results highlight the effectiveness of directly aligning dynamics during offline cross-domain policy adaptation.

**Limitations.** STC needs to train both the forward dynamics model and the inverse policy model. Also, we make several assumptions for the benefit of theoretical analysis, and some of the assumptions may be strong in practice.

## Broader Impact Statement

This work does not introduce obvious ethical concerns or direct negative societal impacts. We do not anticipate significant misuse beyond standard considerations for machine learning systems.

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

## A   Missing Proofs

In this section, we provides detailed proofs for the theoretical results stated in the main text. To enhance readability, we restate each theorem prior to its corresponding proof.

### A.1 Proofs of Theorem 4.4

**Theorem A.1.** Denote the corrected source domain transition dynamics as $\widetilde{P}_{\mathrm{src}}$, then under Assumption 4.1 and 4.2, the deviation between the corrected dynamics and the empirical target domain dynamics $\widehat{P}_{\mathrm{tar}}(\cdot|s,a)$ is bounded:

$$\|\widetilde{P}_{\mathrm{src}}(\cdot|s,a) - \widehat{P}_{\mathrm{tar}}(\cdot|s,a)\| \leq \kappa + \epsilon. \tag{11}$$

*Proof.* Note that $\widetilde{P}_{\mathrm{src}} = \widetilde{P}_{\mathrm{src}}^{\hat{\pi}}, \widehat{P}_{\mathrm{tar}} = \widehat{P}_{\mathrm{tar}}^{\mu_{\mathrm{tar}}}$, where $\hat{\pi}$ is the estimated behavior policy in the corrected data, and $\mu_{\mathrm{tar}}$ is the behavior policy in the target domain dataset. It is easy to find that

$$\|\widetilde{P}_{\mathrm{src}}^{\hat{\pi}}(\cdot|s,a) - P_{\mathrm{tar}}^{\mu_{\mathrm{tar}}}(\cdot|s,a)\| = \left\|\sum_a P_{\mathrm{src}}(s'|s,a)\hat{\pi}(a|s) - \sum_a P_{\mathrm{tar}}(s'|s,a)\mu_{\mathrm{tar}}(a|s)\right\|$$

$$= \left\|\sum_a P_{\mathrm{src}}(s'|s,a)(\hat{\pi}(a|s) - \mu_{\mathrm{tar}}(a|s)) - \sum_a (P_{\mathrm{tar}}(s'|s,a) - P_{\mathrm{src}}(s'|s,a))\mu_{\mathrm{tar}}(a|s)\right\|$$

$$\leq \left\|\sum_a P_{\mathrm{src}}(s'|s,a)(\hat{\pi}(a|s) - \mu_{\mathrm{tar}}(a|s))\right\| + \left\|\sum_a (P_{\mathrm{tar}}(s'|s,a) - P_{\mathrm{src}}(s'|s,a))\mu_{\mathrm{tar}}(a|s)\right\|$$

$$\leq \underbrace{\sum_a \|\hat{\pi}(a|s) - \mu_{\mathrm{tar}}(a|s)\|}_{\leq \kappa} + \underbrace{\|P_{\mathrm{src}}(\cdot|s,a) - P_{\mathrm{tar}}(\cdot|s,a)\|}_{\leq \epsilon}$$

$$\leq \epsilon + \kappa.$$

The above inequalities hold due to Assumption 4.1, 4.2, and the fact that $|P_{\mathrm{src}}(s'|s,a)| \leq 1$ and $\sum_a \mu_{\mathrm{tar}}(a|s) = 1$. $\qquad\square$

### A.2 Proofs of Theorem 4.5

**Theorem A.2.** Given Assumption 4.3 and assume that the source domain rewards are corrected via $\hat{r}(s_{\mathrm{src}}, a_{\mathrm{src}}) = r(s_{\mathrm{src}}, a_{\mathrm{src}}) + \nabla_a r(s_{\mathrm{src}}, a)^\top|_{a=a_{\mathrm{src}}}(\hat{a}_{\mathrm{src}} - a_{\mathrm{src}})$, where $\hat{a}_{\mathrm{src}} \sim \mu_{\mathrm{tar}}(\cdot|s_{\mathrm{src}})$. Then given any $(s,a)$, the deviation of Q-values on the corrected empirical source domain $\widetilde{\mathcal{M}}_{\mathrm{src}}$ and the raw empirical source domain $\widehat{\mathcal{M}}_{\mathrm{src}}$ is bounded:

$$\left|Q_{\widetilde{\mathcal{M}}_{\mathrm{src}}}^\pi(s,a) - Q_{\widehat{\mathcal{M}}_{\mathrm{src}}}^\pi(s,a)\right| \leq \frac{2L_r}{1-\gamma} D_{\mathrm{TV}}(\mu_{\mathrm{src}}\|\mu_{\mathrm{tar}}).$$

*Proof.* Based on the definition of the Q-value, we have

$$Q(s,a) = \mathbb{E}\left[\sum_{t=0}^\infty \gamma^t r(s_t, a_t)\Big| s_0, a_0, \pi\right].$$

Since we only modify the reward signals in the empirical MDP (i.e., $\widehat{\mathcal{M}}_{\mathrm{src}}$ only differs from $\widetilde{\mathcal{M}}_{\mathrm{src}}$ in terms of the reward signals), given the same policy, the induced trajectories are identical. By using Assumption 4.3, we have

$$|Q_{\widetilde{\mathcal{M}}_{\mathrm{src}}}^\pi(s,a) - Q_{\widehat{\mathcal{M}}_{\mathrm{src}}}^\pi(s,a)| = \left|\mathbb{E}_\pi\left[\sum_{t=0}^\infty \gamma^t(\hat{r}(s_t, a_t) - r(s_t, a_t))\Big| s_0 = s, a_0 = a\right]\right|$$

$$\leq \mathbb{E}_\pi\left[\sum_{t=0}^\infty \gamma^t|(\hat{r}(s_t, a_t) - r(s_t, a_t))|\Big| s_0 = s, a_0 = a\right]$$

$$= \mathbb{E}_\pi\left[\sum_{t=0}^\infty \gamma^t|\nabla_a r(\hat{a}_{\mathrm{src}} - a_{\mathrm{src}})|\right].$$

Due to the fact that $\hat{a}_{\mathrm{src}} \sim \mu_{\mathrm{tar}}(\cdot|s_{\mathrm{src}})$, we denote $a_{\mathrm{tar}} = \hat{a}_{\mathrm{src}}$. Then, we have

$$|Q^{\pi}_{\widetilde{\mathcal{M}}_{\mathrm{src}}}(s,a) - Q^{\pi}_{\widehat{\mathcal{M}}_{\mathrm{src}}}(s,a)| \leq \mathbb{E}_{\pi}\left[\sum_{t=0}^{\infty}\gamma^t|\nabla_a r|\|a_{\mathrm{tar}} - a_{\mathrm{src}}\|\right] \leq \mathbb{E}_{\pi}\left[\sum_{t=0}^{\infty}\gamma^t L_r\|a_{\mathrm{tar}} - a_{\mathrm{src}}\|\right]$$

$$\leq \sum_{t=0}^{\infty}\gamma^t L_r \sum \|a_{\mathrm{tar}} - a_{\mathrm{src}}\|$$

$$= \frac{L_r}{1-\gamma}\sum \|a_{\mathrm{tar}} - a_{\mathrm{src}}\| = \frac{2L_r}{1-\gamma}D_{\mathrm{TV}}(\mu_{\mathrm{tar}}\|\mu_{\mathrm{src}}).$$

$\square$

### A.3 Proofs of Theorem 4.6

**Theorem A.3** (Finite data bound). *Denote $\widetilde{\mathcal{M}}_{\mathrm{src}}$ as the corrected empirical source domain MDP, $n$ is the size of the target domain dataset, $C_1 = \frac{\gamma r_{\max}|\mathcal{S}|}{\sqrt{2}(1-\gamma)^2}$, $C_2 = |\mathcal{S} \times \mathcal{A} \times \mathcal{S}|$. Then under Assumption 4.1-4.3, for any policy $\pi$, the following bound holds with probability at least $1 - \delta$:*

$$J_{\widetilde{\mathcal{M}}_{\mathrm{src}}}(\pi) - J_{\mathcal{M}_{\mathrm{tar}}}(\pi) \geq -\frac{\gamma r_{\max}(\kappa + \epsilon)}{(1-\gamma)^2} - C_1\sqrt{\frac{1}{n}\ln\frac{2C_2}{\delta}}.$$

*Proof.* We write the return of a policy in the MDP $\mathcal{M}$ with the following form:

$$J_{\mathcal{M}}(\pi) = \mathbb{E}_{s,a\sim\rho^{\pi}_{\mathcal{M}}}[r(s,a)]. \tag{12}$$

Note that $\mathcal{M}_{\mathrm{tar}}$ denotes the true target domain MDP rather than the empirical target domain MDP $\widehat{\mathcal{M}}_{\mathrm{tar}}$. We then decompose the return difference into the following form:

$$J_{\widetilde{\mathcal{M}}_{\mathrm{src}}}(\pi) - J_{\mathcal{M}_{\mathrm{tar}}}(\pi) = \underbrace{J_{\widetilde{\mathcal{M}}_{\mathrm{src}}}(\pi) - J_{\widehat{\mathcal{M}}_{\mathrm{tar}}}(\pi)}_{:=(\mathrm{i})} + \underbrace{J_{\widehat{\mathcal{M}}_{\mathrm{tar}}}(\pi) - J_{\mathcal{M}_{\mathrm{tar}}}(\pi)}_{:=(\mathrm{ii})}. \tag{13}$$

To show the desired conclusion, we need the following lemma.

**Lemma A.1** (Telescoping lemma). *Denote $\mathcal{M}_1 = (\mathcal{S}, \mathcal{A}, P_1, r, \gamma)$ and $\mathcal{M}_2 = (\mathcal{S}, \mathcal{A}, P_2, r, \gamma)$ as two MDPs that only differ in their transition dynamics. Then for any policy $\pi$, we have*

$$J_{\mathcal{M}_1}(\pi) - J_{\mathcal{M}_2}(\pi) = \frac{\gamma}{1-\gamma}\mathbb{E}_{\rho^{\pi}_{\mathcal{M}_1}(s,a)}\left[\mathbb{E}_{s'\sim P_1}[V^{\pi}_{\mathcal{M}_2}(s')] - \mathbb{E}_{s'\sim P_2}[V^{\pi}_{\mathcal{M}_2}(s')]\right]. \tag{14}$$

The proof of the above lemma can be found in Luo et al. (2019).

For term (i), we use the above lemma and have

$$\left|J_{\widetilde{\mathcal{M}}_{\mathrm{src}}}(\pi) - J_{\widehat{\mathcal{M}}_{\mathrm{tar}}}(\pi)\right| = \left|\frac{\gamma}{1-\gamma}\mathbb{E}_{\rho^{\pi}_{\widetilde{\mathcal{M}}_{\mathrm{src}}}}\left[\mathbb{E}_{s'\sim\widetilde{P}_{\mathrm{src}}}[V^{\pi}_{\widehat{\mathcal{M}}_{\mathrm{tar}}}(s')] - \mathbb{E}_{s'\sim\widehat{P}_{\mathrm{tar}}}[V^{\pi}_{\widehat{\mathcal{M}}_{\mathrm{tar}}}(s')]\right]\right|$$

$$\leq \frac{\gamma}{1-\gamma}\mathbb{E}_{\rho^{\pi}_{\widetilde{\mathcal{M}}_{\mathrm{src}}}}\left|\mathbb{E}_{s'\sim\widetilde{P}_{\mathrm{src}}}[V^{\pi}_{\widehat{\mathcal{M}}_{\mathrm{tar}}}(s')] - \mathbb{E}_{s'\sim\widehat{P}_{\mathrm{tar}}}[V^{\pi}_{\widehat{\mathcal{M}}_{\mathrm{tar}}}(s')]\right|$$

$$\leq \frac{\gamma}{1-\gamma}\mathbb{E}_{\rho^{\pi}_{\widetilde{\mathcal{M}}_{\mathrm{src}}}}\left|\sum_{s'}\left[\widetilde{P}_{\mathrm{src}}(s'|s,a) - \widehat{P}_{\mathrm{tar}}(s'|s,a)\right]V^{\pi}_{\widehat{\mathcal{M}}_{\mathrm{tar}}}(s')\right|$$

$$\leq \frac{\gamma}{1-\gamma}\frac{r_{\max}}{1-\gamma}\mathbb{E}_{\rho^{\pi}_{\widetilde{\mathcal{M}}_{\mathrm{src}}}}\left|\sum_{s'}\left[\widetilde{P}_{\mathrm{src}}(s'|s,a) - \widehat{P}_{\mathrm{tar}}(s'|s,a)\right]\right|$$

$$\leq \frac{\gamma r_{\max}}{(1-\gamma)^2}\mathbb{E}_{\rho^{\pi}_{\widetilde{\mathcal{M}}_{\mathrm{src}}}}\sum_{s'}\left|\widetilde{P}_{\mathrm{src}}(s'|s,a) - \widehat{P}_{\mathrm{tar}}(s'|s,a)\right|$$

$$= \frac{\gamma r_{\max}}{(1-\gamma)^2}\mathbb{E}_{\rho^{\pi}_{\widetilde{\mathcal{M}}_{\mathrm{src}}}}\left\|\widetilde{P}_{\mathrm{src}}(s'|s,a) - \widehat{P}_{\mathrm{tar}}(s'|s,a)\right\|_1$$

$$\leq \frac{\gamma r_{\max}}{(1-\gamma)^2}(\kappa + \epsilon),$$

where the last inequality holds due to Theorem 4.4. We also use the fact that $V_{\mathcal{M}}^{\pi}(s) \leq \frac{r_{\max}}{1-\gamma}$.

Then we have,

$$J_{\widetilde{\mathcal{M}}_{\mathrm{src}}}(\pi) - J_{\widehat{\mathcal{M}}_{\mathrm{tar}}}(\pi) \geq -\frac{\gamma r_{\max}}{(1-\gamma)^2}(\kappa + \epsilon),$$

For term (ii), we also use the above lemma and have

$$
\begin{aligned}
J_{\widehat{\mathcal{M}}_{\mathrm{tar}}}(\pi) - J_{\mathcal{M}_{\mathrm{tar}}}(\pi) &= \frac{\gamma}{1-\gamma} \mathbb{E}_{\rho_{\widehat{\mathcal{M}}_{\mathrm{tar}}}^{\pi}} \left[ \mathbb{E}_{s' \sim \widehat{P}_{\mathrm{tar}}}[V_{\mathcal{M}_{\mathrm{tar}}}^{\pi}(s')] - \mathbb{E}_{s' \sim P_{\mathrm{tar}}}[V_{\mathcal{M}_{\mathrm{tar}}}^{\pi}(s')] \right] \\
&= \frac{\gamma}{1-\gamma} \mathbb{E}_{\rho_{\widehat{\mathcal{M}}_{\mathrm{tar}}}^{\pi}} \left[ \sum_{s'} \left[ \widehat{P}_{\mathrm{tar}}(s'|s,a) - P_{\mathrm{tar}}(s'|s,a) \right] V_{\mathcal{M}_{\mathrm{tar}}}^{\pi}(s') \right] \\
&\geq -\frac{\gamma}{1-\gamma} \mathbb{E}_{\rho_{\widehat{\mathcal{M}}_{\mathrm{tar}}}^{\pi}} \left[ \sum_{s'} \left| \widehat{P}_{\mathrm{tar}}(s'|s,a) - P_{\mathrm{tar}}(s'|s,a) \right| V_{\mathcal{M}_{\mathrm{tar}}}^{\pi}(s') \right] \\
&\geq -\frac{\gamma}{1-\gamma} \frac{r_{\max}}{1-\gamma} \mathbb{E}_{\rho_{\widehat{\mathcal{M}}_{\mathrm{tar}}}^{\pi}} \left[ \sum_{s'} \left| \widehat{P}_{\mathrm{tar}}(s'|s,a) - P_{\mathrm{tar}}(s'|s,a) \right| \right] \\
&= -\frac{\gamma r_{\max}}{(1-\gamma)^2} \mathbb{E}_{\rho_{\widehat{\mathcal{M}}_{\mathrm{tar}}}^{\pi}} \left\| \widehat{P}_{\mathrm{tar}}(s'|s,a) - P_{\mathrm{tar}}(s'|s,a) \right\|_1.
\end{aligned}
$$

Note that $P_{\mathrm{tar}}(s'|s,a)$ and $\widehat{P}_{\mathrm{tar}}(s'|s,a)$ returns the probability of the next state under a given state-action pair $(s,a)$. Then under a fixed $(s,a)$, we have

$$\left\| \widehat{P}_{\mathrm{tar}}(s'|s,a) - P_{\mathrm{tar}}(s'|s,a) \right\|_1 \leq |\mathcal{S}| \left\| \widehat{P}_{\mathrm{tar}}(s'|s,a) - P_{\mathrm{tar}}(s'|s,a) \right\|_{\infty}.$$

For $\left\| \widehat{P}_{\mathrm{tar}}(s'|s,a) - P_{\mathrm{tar}}(s'|s,a) \right\|_{\infty}$, we bound it with the Hoeffding's inequality and union bound,

$$\mathbb{P}\left( \left| \widehat{P}_{\mathrm{tar}}(s'|s,a) - P_{\mathrm{tar}}(s'|s,a) \right| > \epsilon \right) \leq 2|\mathcal{S} \times \mathcal{A} \times \mathcal{S}| \exp(-2n\epsilon^2),$$

where $n$ is the size of the target domain offline dataset, $\mathbb{P}(\cdot)$ is the probability measure. To make the above probability less than $\delta$, we have

$$2|\mathcal{S} \times \mathcal{A} \times \mathcal{S}| \exp(-2n\epsilon^2) < \delta$$

$$\Rightarrow \epsilon > \sqrt{\frac{1}{2n} \ln \frac{2|\mathcal{S} \times \mathcal{A} \times \mathcal{S}|}{\delta}}.$$

That said,

$$\mathbb{P}\left( \left| \widehat{P}_{\mathrm{tar}}(s'|s,a) - P_{\mathrm{tar}}(s'|s,a) \right| > \sqrt{\frac{1}{2n} \ln \frac{2|\mathcal{S} \times \mathcal{A} \times \mathcal{S}|}{\delta}} \right) < \delta,$$

Therefore, with probability at least $1 - \delta$, we have

$$\left\| \widehat{P}_{\mathrm{tar}}(s'|s,a) - P_{\mathrm{tar}}(s'|s,a) \right\|_{\infty} \leq \sqrt{\frac{1}{2n} \ln \frac{2|\mathcal{S} \times \mathcal{A} \times \mathcal{S}|}{\delta}}.$$

We then bound term (ii) below,

$$
\begin{aligned}
J_{\widehat{\mathcal{M}}_{\mathrm{tar}}}(\pi) - J_{\mathcal{M}_{\mathrm{tar}}}(\pi) &= -\frac{\gamma r_{\max}}{(1-\gamma)^2} \mathbb{E}_{\rho_{\widehat{\mathcal{M}}_{\mathrm{tar}}}^{\pi}} \left\| \widehat{P}_{\mathrm{tar}}(s'|s,a) - P_{\mathrm{tar}}(s'|s,a) \right\|_1 \\
&\geq -\frac{\gamma r_{\max}}{(1-\gamma)^2} |\mathcal{S}| \sqrt{\frac{1}{2n} \ln \frac{2|\mathcal{S} \times \mathcal{A} \times \mathcal{S}|}{\delta}} \\
&= -\frac{\gamma r_{\max} |\mathcal{S}|}{\sqrt{2}(1-\gamma)^2} \sqrt{\frac{1}{n} \ln \frac{2|\mathcal{S} \times \mathcal{A} \times \mathcal{S}|}{\delta}}.
\end{aligned}
$$

By denoting $C_1 = \frac{\gamma r_{\max} |\mathcal{S}|}{\sqrt{2}(1-\gamma)^2}, C_2 = |\mathcal{S} \times \mathcal{A} \times \mathcal{S}|$ and combining the bounds of term (i) and term (ii), the conclusion follows as is. $\qquad \square$

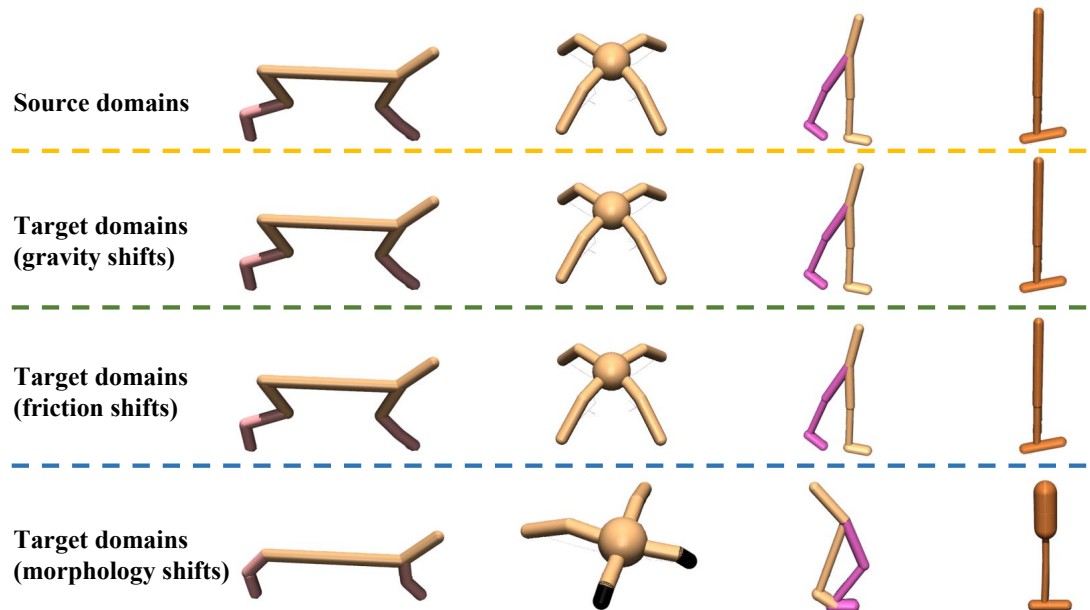

Figure 4: **Illustration of the adopted environments.** Target domain robots differ from source domain robots (top) by gravity shifts (second row), friction shifts (third row), or morphology shifts (bottom).

## B  Environment Setting

In this section, we introduce the experimental setup used to evaluate STC. We first describe the datasets, followed by details of the three domain shift settings that we adopt.

### B.1  Datasets

We directly adopt the MuJoCo datasets from D4RL Fu et al. (2020) as our source domain datasets. These datasets are collected through interactions with continuous control environments in Gym Brockman et al. (2016), simulated using MuJoCo Todorov et al. (2012). We select four representative tasks: *HalfCheetah, Hopper, Walker2d,* and *Ant,* and utilize datasets of three different quality levels: *medium, medium-replay,* and *medium-expert.*

For the target domain datasets, we consider three types of dynamics shifts: gravity shift, friction shift, and morphology shift, across four MuJoCo tasks (Ant, Hopper, HalfCheetah, Walker2d) from the ODRL benchmark Lyu et al. (2024b). Figure 4 presents visual comparisons between the source and target domain agents. Detailed configurations for each task are provided in later sections. We use datasets of three quality levels: *medium, medium-expert,* and *expert.* The expert dataset is collected using a SAC policy trained for 1 million steps. The medium dataset is generated using checkpoints with performance around one-half or one-third of the expert policy. The medium-expert dataset is composed of 2 trajectories from the medium set and 3 from the expert set. Due to our focus on settings with restricted access to target domain data in main experiments, the target domain dataset contains approximately 5,000 transitions.

### B.2  Metrics

To ensure that the results are interpretable across different tasks, we follow ODRL Lyu et al. (2024b) and adopt the normalized score (NS) in the target domain as the evaluation metric:

$$\text{NS} = \frac{J - J_r}{J_e - J_r} \times 100, \tag{15}$$

Table 7: **The Reference min scores $J_r$ and max scores $J_e$ for tasks under dynamics shifts.** The scores are used to compute normalized scores in the target domain.

| Task Name | Dynamics shift type | $J_r$ | $J_e$ |
|---|---|---|---|
| halfcheetah | gravity | -280.18 | 9509.15 |
| halfcheetah | morphology | -280.18 | 12135.00 |
| halfcheetah | friction | -280.18 | 7357.07 |
| hopper | gravity | -26.34 | 3234.30 |
| hopper | morphology | -26.34 | 3234.30 |
| hopper | friction | -26.34 | 3234.30 |
| walker2d | gravity | 10.08 | 5194.71 |
| walker2d | morphology | 10.08 | 4592.30 |
| walker2d | friction | 10.08 | 4229.35 |
| ant | gravity | -325.60 | 4317.07 |
| ant | morphology | -325.60 | 5139.83 |
| ant | friction | -325.60 | 8301.34 |

where $J$, $J_e$, and $J_r$ denote the returns of the learned policy, the expert policy and the random policy in the target domain, respectively. We list the reference scores of $J_r$ and $J_e$ under different dynamics shift scenarios in Table 7.

### B.3 Gravity Shift Tasks

Gravity shifts are introduced by editing the environment XML files, where the gravitational acceleration in the target domain is set to 50% of that in the source domain, with the force direction preserved.

***halfcheetah / hopper / walker2d / ant-gravity:*** The modifications of the XML file gives:

```
# gravity
<option gravity="0 0 -4.905" timestep="0.01"/>
```

### B.4 Morphology Shift Tasks

The morphology shift modifies the size of specific limbs or torsos of the simulated robot in the target domain. We modify the XML files of each environment to introduce task-specific changes, as detailed below.

***halfcheetah-morph:*** The sizes of the back thigh and the forward thigh of the Cheetah robot are revised as below:

```
<geom fromto="0 0 0 0.08 0 -0.08" name="bthigh" size="0.046" type="capsule"/>
<body name="bshin" pos="0.08 0 -0.08">
<geom fromto="0 0 0 -.13 0 -.15" name="bshin" rgba="0.9 0.6 0.6 1" size="0.046"
    type="capsule"/>
<body name="bfoot" pos="-.13 0 -.15">
<geom fromto="0 0 0 -0.07 0 -0.08" name="fthigh" size="0.046" type="capsule"/>
<body name="fshin" pos="-0.07 0 -0.08">
<geom fromto="0 0 0 .11 0 -.13" name="fshin" rgba="0.9 0.6 0.6 1" size="0.046"
    type="capsule"/>
<body name="ffoot" pos=".11 0 -.13">
```

***hopper-morph:*** The foot size is revised to be 0.6 times of that in the source domain:

```
<geom friction="2.0" fromto="-0.078 0 0.1 0.156 0 0.1" name="foot_geom" size="
    0.036" type="capsule"/>
```

***walker2d-morph:*** The leg size of the robot is revised to be 0.5 times of that in the source domain:

```
<geom friction="0.9" fromto="0 0 1.05 0 0 0.35" name="thigh_geom" size="0.05" type
    ="capsule"/>
<joint axis="0 -1 0" name="leg_joint" pos="0 0 0.35" range="-150 0" type="hinge"/>
<geom friction="0.9" fromto="0 0 0.35 0 0 0.1" name="leg_geom" size="0.04" type="
    capsule"/>
<geom friction="0.9" fromto="0 0 1.05 0 0 0.35" name="thigh_left_geom" rgba=".7 .3
     .6 1" size="0.05" type="capsule"/>
<joint axis="0 -1 0" name="leg_left_joint" pos="0 0 0.35" range="-150 0" type="
    hinge"/>
<geom friction="0.9" fromto="0 0 0.35 0 0 0.1" name="leg_left_geom" rgba=".7 .3 .6
     1" size="0.04" type="capsule"/>
```

***ant-morph:*** The sizes of the front two legs are revised to be 0.5 times of those in the source domain:

```
<geom fromto="0.0 0.0 0.0 0.2 0.2 0.0" name="left_ankle_geom" size="0.08" type="
    capsule"/>
<geom fromto="0.0 0.0 0.0 -0.2 0.2 0.0" name="right_ankle_geom" size="0.08" type="
    capsule"/>
```

### B.5 Friction Shift Tasks

The friction shift is introduced by modifying the friction attributes in each environment, setting them to 0.5 times the values used in the source domain.

***halfcheetah / hopper / walker2d / ant-friction:*** The corresponding XML files are modified accordingly, as detailed below:

```
<geom conaffinity="0" condim="3" density="5.0" friction="0.5 0.25 0.25" margin="
    0.01" rgba="0.8 0.6 0.4 1"/>
```

## C  Algorithmic Implementation

In this section, we present the implementation details of our proposed method, STC, as well as all baseline approaches considered in this paper. In addition, we report the corresponding hyperparameter configurations for each method.

### C.1  Implementation Details

**IQL:** Implicit Q-Learning (IQL) Kostrikov et al. (2022) is a popular offline RL algorithm that learns policies solely from in-sample data without querying out-of-distribution samples. However, we observe that training IQL only on the target domain dataset yields suboptimal policies. To address this, we modify IQL to jointly leverage both source and target domain data. The state value function in IQL is trained via expectile regression:

$$\mathcal{L}_V = \mathbb{E}_{(s,a)\sim D_{\mathrm{src}}\cup D_{\mathrm{tar}}}[L_2^\tau(Q_{\theta'}(s,a) - V_\psi(s))], \tag{16}$$

where $L_2^\tau(u) = |\tau - \mathbf{1}(u < 0)|u^2$ and $\theta'$ denotes target network parameters. The Q-function update minimizes:

$$\mathcal{L}_Q = \mathbb{E}_{(s,a,r,s')\sim D_{\mathrm{src}}\cup D_{\mathrm{tar}}}[(r(s,a) + \gamma V_\psi(s') - Q_\theta(s,a))^2]. \tag{17}$$

Then the policy is updated by:

$$\mathcal{L}_{\mathrm{actor}} = \mathbb{E}_{(s,a) \sim D_{\mathrm{src}} \cup D_{\mathrm{tar}}} \left[ \exp(\beta_{\mathrm{IQL}} A(s,a)) \log \pi_\phi(a|s) \right], \tag{18}$$

where $A(s,a) = Q(s,a) - V(s)$ is the advantage function, and $\beta_{\mathrm{IQL}}$ is the inverse temperature coefficient. We implement IQL based on the official codebase[1] and adopt symmetric sampling when sampling data from the source dataset and target dataset.

**DARA:** DARA Liu et al. (2022) is the offline version of DARC Eysenbach et al. (2021). It also trains two domain classifiers, $q_{\theta_{\mathrm{SAS}}}(\mathrm{target}|s_t, a_t, s_{t+1})$ and $q_{\theta_{\mathrm{SA}}}(\mathrm{target}|s_t, a_t)$, with objectives:

$$\mathcal{L}(\theta_{\mathrm{SAS}}) = \mathbb{E}_{D_{\mathrm{tar}}} \left[ \log q_{\theta_{\mathrm{SAS}}}(\mathrm{target}|s_t, a_t, s_{t+1}) \right] + \mathbb{E}_{D_{\mathrm{src}}} \left[ \log(1 - q_{\theta_{\mathrm{SAS}}}(\mathrm{target}|s_t, a_t, s_{t+1})) \right],$$
$$\mathcal{L}(\theta_{\mathrm{SA}}) = \mathbb{E}_{D_{\mathrm{tar}}} \left[ \log q_{\theta_{\mathrm{SA}}}(\mathrm{target}|s_t, a_t) \right] + \mathbb{E}_{D_{\mathrm{src}}} \left[ \log(1 - q_{\theta_{\mathrm{SA}}}(\mathrm{target}|s_t, a_t)) \right].$$

The classifiers are employed to estimate the dynamics gap $\log \frac{P_{\mathcal{M}_{\mathrm{tar}}}(s_{t+1}|s_t, a_t)}{P_{\mathcal{M}_{\mathrm{src}}}(s_{t+1}|s_t, a_t)}$ between the source and target domains, which is used to adjust the source domain rewards:

$$\hat{r}_{\mathrm{DARA}} = r - \lambda \times \delta_r,$$
$$\delta_r(s_t, a_t) = -\log \left( \frac{q_{\theta_{\mathrm{SAS}}}(\mathrm{target}|s_t, a_t, s_{t+1}) \cdot q_{\theta_{\mathrm{SA}}}(\mathrm{source}|s_t, a_t)}{q_{\theta_{\mathrm{SAS}}}(\mathrm{source}|s_t, a_t, s_{t+1}) \cdot q_{\theta_{\mathrm{SA}}}(\mathrm{target}|s_t, a_t)} \right), \tag{19}$$

where $\lambda$ controls the penalty strength. We empirically find that setting $\lambda = 1$ or higher often degrades performance, so we use $\lambda = 0.1$ by default. Our implementation follows the attached code on its OpenReview page[2], and use IQL as the base algorithm for DARA to maintain consistency with other methods. To ensure training stability, we clip the penalty term within $[-10, 10]$.

**BOSA:** To tackle cross-domain offline RL, BOSA Liu et al. (2024) introduces two support constraints: one for policy learning to alleviate the out-of-distribution (OOD) state-action problem, and another for value learning to handle the OOD dynamics issue. Specifically, the critics of BOSA are trained using:

$$\mathcal{L}_{\mathrm{critic}} = \mathbb{E}_{(s,a) \sim D_{\mathrm{src}}}[Q_{\theta_i}(s,a)] + \mathbb{E}_{\substack{(s,a,r,s') \sim D_{\mathrm{src}} \cup D_{\mathrm{tar}}, \\ a' \sim \pi_\phi(\cdot|s)}} \left[ \mathbf{1}(\hat{P}_{\mathrm{tar}}(s'|s,a) > \epsilon)(Q_{\theta_i}(s,a) - y)^2 \right], \tag{20}$$

where $\mathbf{1}(\cdot)$ denotes an indicator function, $\hat{P}_{\mathrm{tar}}(s'|s,a) = \arg\max \mathbb{E}_{(s,a,s') \sim D_{\mathrm{tar}}}[\log \hat{P}_{\mathrm{tar}}(s'|s,a)]$ is the estimated transition model for the target domain, and $\epsilon$ is a filtering threshold. The index $i \in \{1, 2\}$ indicates two critics. The actor is trained via a supported policy optimization objective:

$$\mathcal{L}_{\mathrm{actor}} = \mathbb{E}_{s \sim D_{\mathrm{src}} \cup D_{\mathrm{tar}}, a \sim \pi_\phi(s)}[Q_{\theta_i}(s,a)], \quad \text{s.t.} \ \mathbb{E}_{s \sim D_{\mathrm{src}} \cup D_{\mathrm{tar}}}[\hat{\pi}_{\phi_{\mathrm{mix}}}(\pi_\phi(s)|s)] > \epsilon', \tag{21}$$

where $\hat{\pi}_{\phi_{\mathrm{mix}}}$ is a learned behavior model over the combined dataset, and $\epsilon'$ is a predefined threshold. BOSA models both the transition dynamics in the target domain and the behavior policy of the mixed dataset using CVAE. As there is no official implementation, we use the BOSA's implementation by ODRL Lyu et al. (2024b), which adopts SPOT Wu et al. (2022) as its backbone. In our experiments, BOSA is trained with 1M gradient steps using samples drawn from both source and target domain datasets.

**SRPO:** SRPO Xue et al. (2024) formulates policy learning as a constrained optimization problem:

$$\max_\pi \mathbb{E}_{s_t, a_t \sim \tau_\pi} \left[ \sum_{t=0}^{\infty} \gamma^t r(s_t, a_t) \right] \quad \text{s.t.} \quad D_{\mathrm{KL}}(d_\pi(\cdot) \| \zeta(\cdot)) < \epsilon, \tag{22}$$

where $\tau_\pi$ denotes the trajectory under policy $\pi$, $d_\pi(\cdot)$ is the corresponding stationary state distribution, and $\zeta(\cdot)$ represents the optimal state distribution under alternate dynamics. The problem then can be transformed into the unconstrained optimization problem via Lagrange multipliers, where the logarithm of

---

[1] https://github.com/ikostrikov/implicit_q_learning
[2] https://openreview.net/forum?id=9SDQB3b68K

probability density ratio $\lambda \log \frac{\zeta(s_t)}{d_\pi(s_t)}$ is added to the vanilla reward term. In practice, SRPO samples a batch of $N$ transitions from the combined dataset $D_{\mathrm{src}} \cup D_{\mathrm{tar}}$, ranks them by estimated state value, and labels the top $\rho N$ transitions as *real*, with the remaining marked as *fake*. A discriminator $D_\delta(s)$ is trained to classify them, and the reward is modified as:

$$\hat{r}_{\mathrm{SRPO}} = r + \lambda \times \frac{D_\delta(s)}{1 - D_\delta(s)}, \tag{23}$$

where $\lambda$ is a scaling coefficient. Following the original setup, we set $\rho = 0.5$ in all experiments. As no official implementation is available, we reproduce SRPO based on the descriptions in the paper.

**IGDF:** IGDF Wen et al. (2024) leverages contrastive learning to capture the dynamics discrepancy between source and target domains. A score function $h(\cdot)$ is trained using positive samples $(s, a, s'_{\mathrm{tar}}) \sim D_{\mathrm{tar}}$ and negative samples constructed by pairing $(s, a) \sim D_{\mathrm{tar}}$ with $s'_{\mathrm{src}} \sim D_{\mathrm{src}}$, forming $(s, a, s'_{\mathrm{src}})$. The training objective is:

$$\mathcal{L}_{\mathrm{contrastive}} = -\mathbb{E}_{(s,a,s'_{\mathrm{tar}})}\mathbb{E}_{S'^-}\left[\log \frac{h(s, a, s'_{\mathrm{tar}})}{\sum_{s' \sim S'^- \cup \{s'_{\mathrm{tar}}\}} h(s, a, s')}\right], \tag{24}$$

where $S'^-$ denotes a set of negative next states. To parameterize $h$, IGDF employs two neural networks $\phi(s, a)$ and $\psi(s')$ to encode state-action and state representations, respectively, and defines the score function as:

$$h(s, a, s') = \exp\left(\phi(s, a)^T \psi(s')\right). \tag{25}$$

Using the learned score function, IGDF selectively incorporates source domain samples into critic training by filtering transitions with low dynamics consistency:

$$\begin{aligned}
\mathcal{L}_{\mathrm{critic}} = &\frac{1}{2}\mathbb{E}_{D_{\mathrm{tar}}}\left[(Q_\theta - \mathcal{T}Q_\theta)^2\right] \\
&+ \frac{1}{2}\alpha \cdot h(s, a, s')\mathbb{E}_{(s,a,s') \sim D_{\mathrm{src}}}\left[\mathbf{1}(h(s, a, s') > h_{\xi\%})(Q_\theta - \mathcal{T}Q_\theta)^2\right],
\end{aligned} \tag{26}$$

where $\alpha$ controls the influence of the source domain loss, and $\xi$ denotes the percentile threshold for filtering source domain samples. We adopt the official implementation[3] to run IGDF and use IQL as the backbone throughout all experiments.

**OTDF** OTDF Lyu et al. (2025) aims to mitigate domain shift by selectively utilizing source domain data aligned with the target domain via optimal transport (OT). It first solves an OT problem to align source and target datasets, computing per-sample deviations $\{d_t\}_{t=1}^{|D_{\mathrm{src}}|}$ that quantify alignment quality with:

$$d(u_t) = -\sum_{t'=1}^{|D_{\mathrm{tar}}|} C(u_t, u'_{t'})\mu^*_{t,t'}, \quad u_t = (s^t_{\mathrm{src}}, a^t_{\mathrm{src}}, (s'_{\mathrm{src}})^t) \sim D_{\mathrm{src}}. \tag{27}$$

These deviations are appended to source transitions, forming an augmented dataset $\hat{D}_{\mathrm{src}} = \{(s_t, a_t, s'_t, r_t, d_t)\}_{t=1}^{|D_{\mathrm{src}}|}$. Then a CVAE policy is trained on $D_{\mathrm{tar}}$ to model its behavior policy, which is later used for policy regularization. At each iteration, mini-batches are sampled from both $D_{\mathrm{tar}}$ and $\hat{D}_{\mathrm{src}}$. The top $\xi\%$ of source samples—those best aligned with the target—are retained, and their critic losses are weighted by the normalized deviations $\hat{d}_i = \frac{d_i - \max d_i}{\max d_i - \min d_i}, i \in \{1, 2, \ldots, N\}$. The state-action value function $Q_\theta$ is optimized via:

$$\mathcal{L}_Q = \mathbb{E}_{D_{\mathrm{tar}}}[(Q_\theta - y)^2] + \mathbb{E}_{(s,a,s',r,d) \sim \hat{D}_{\mathrm{src}}}[\exp(\hat{d}) \cdot \mathbf{1}(d > d_{\xi\%})(Q_\theta - y)^2], \tag{28}$$

where $y = r + \gamma V_\psi(s')$. The policy is optimized using advantage-weighted regression (AWR) and a regularization term based on CVAE-decoded actions:

$$\mathcal{L}_\pi = \mathbb{E}_{(s,a) \sim \hat{D}_{\mathrm{src}} \cup D_{\mathrm{tar}}}[\exp(\beta_{\mathrm{IQL}} \cdot A) \log \pi_\phi(a|s)] - \beta \cdot \mathbb{E}_{s \sim \hat{D}_{\mathrm{src}} \cup D_{\mathrm{tar}}}\left[\log \sum_{i=1}^M \hat{\pi}^i_{\mathrm{tar}}(\pi(\cdot|s)|s)\right], \tag{29}$$

where $A$ is the advantage. We run OTDF by following its official codebase[4], with IQL as the backbone.

---

[3]https://github.com/BattleWen/IGDF
[4]https://github.com/dmksjfl/OTDF

**STC**  Different from the aforementioned methods, STC mitigates the dynamics gap by selectively correcting source domain transitions. As this work focuses on cross-domain offline RL, we use the target domain offline dataset to pretrain the inverse policy model, forward dynamics model, and reward model for 50,000 steps via Equation 1, 5 and 3. After the pretraining phase, each training iteration begins by sampling mini-batches from both the source and target domains. For source domain transitions, we first perform action correction using the inverse policy model as defined in Equation 2, and estimate the corresponding corrected rewards via a first-order Taylor approximation (Equation 4). To enhance the reliability of the correction process, we compute the dynamics discrepancy between the corrected and target transitions using Equation 6, and selectively retain those transitions that better conform to the target dynamics based on a thresholding criterion (Equation 7), where the correction threshold $\lambda$ serves as a hyperparameter. Subsequently, the value function is updated by minimizing the temporal-difference (TD) error. We adopt a Q-value-weighted behavior cloning term for the policy optimization objective (Equation 10), which encourages the policy to maximize the estimated Q-values while remaining close to the behavior policy. We implement STC based on the IQL framework, and provide its detailed pseudocode in Appendix D.

## C.2  Hyperparameter Setup

We summarize the specific hyperparameter configurations for each baseline method and STC in Table 8. For IQL, DARA, and BOSA, we employ a unified and fixed set of hyperparameters across all tasks. For SRPO, we report the best performance by sweeping the reward coefficient $\lambda$ over the range $\{0.1, 0.3\}$. For IGDF, we set the data selection ratio $\xi\%$ to 75% and additionally tune the representation dimension over $\{16, 64\}$, reporting the best-performing configuration. For OTDF, we adopt the hyperparameter settings provided in the official implementation, using a fixed $\xi\% = 80\%$ and setting the policy coefficient $\beta$ to either 0.1 or 0.5 depending on the specific task. For STC, we fix the reward gradient coefficient $\alpha$ at 0.5, sweep the correction threshold $\lambda$ over $\{1.0, 5.0\}$, and tune the Q-weighted loss coefficient $\beta$ in the range $\{0.5, 5.0\}$, reporting the best result for each environment.

---

**Algorithm 1** Selective Transition Correction (STC)

---

**Input:** Source domain dataset $D_{\text{src}}$, target domain dataset $D_{\text{tar}}$, batch size $N$
**Initialize:** policy $\pi_\phi$, value function $Q_\theta$, inverse policy model $f_\zeta^{\text{inv}}$, forward dynamics model $f_\xi^{\text{fwd}}$, reward model $r_\nu$, coefficients $\lambda, \alpha, \beta$

1: Train the inverse policy model $f_\zeta^{\text{inv}}$ with $D_{\text{tar}}$  **via Equation equation 1**
2: Train the forward dynamics model $f_\xi^{\text{fwd}}$ with $D_{\text{tar}}$  **via Equation equation 5**
3: Train the reward model $r_\nu$ with $D_{\text{tar}}$  **via Equation equation 3**
4: **for** $i = 1, 2, \ldots$ **do**
5:    Sample a mini-batch $b_{\text{src}} := \{(s_{\text{src}}, a_{\text{src}}, s'_{\text{src}}, r_{\text{src}})\}$ with size $N/2$ from $D_{\text{src}}$
6:    Sample a mini-batch $b_{\text{tar}} = \{(s_{\text{tar}}, a_{\text{tar}}, s'_{\text{tar}}, r_{\text{tar}})\}$ with size $N/2$ from $D_{\text{tar}}$
7:    Modify both the actions and rewards of source transitions to form $\widetilde{b}_{\text{src}} = \{(s_{\text{src}}, \hat{a}_{\text{src}}, s'_{\text{src}}, \hat{r}_{\text{src}})\}$ via:

$$\hat{a}_{\text{src}} = f_{\text{inv}}(s_{\text{src}}, s'_{\text{src}}), \quad \hat{r}_{\text{src}} = r_{\text{src}} + \alpha \cdot \nabla_a r(s_{\text{src}}, a)^\top|_{a=a_{\text{src}}}(\hat{a}_{\text{src}} - a_{\text{src}})$$

8:    Compute *dynamics discrepancies* **with Equation equation 6**
9:    Select corrected source transitions **with Equation equation 7**
10:   Optimize the value function $Q_\theta$ **with Equation equation 9**
11:   Optimize the policy $\pi_\phi$ **with Equation equation 10**
12: **end for**

---

# D  Pseudocode and Details of STC

In this section, we provide the detailed pseudocode of STC, as shown in Algorithm 1.

Table 8: **Hyperparameter setup for STC and baselines.**

| Hyperparameter | Value |
|---|---|
| **Shared** | |
| Actor network | (256, 256) |
| Critic network | (256, 256) |
| Learning rate | $3 \times 10^{-4}$ |
| Optimizer | Adam |
| Discount factor | 0.99 |
| Nonlinearity | ReLU |
| Target update rate | $5 \times 10^{-3}$ |
| Source domain Batch size | 128 |
| Target domain Batch size | 128 |
| **IQL** | |
| Temperature coefficient | 0.2 |
| Maximum log std | 2 |
| Minimum log std | $-20$ |
| Inverse temperature parameter $\beta_{\text{IQL}}$ | 3.0 |
| Expectile parameter $\tau$ | 0.7 |
| **DARA** | |
| Temperature coefficient | 0.2 |
| Classifier network | (256, 256) |
| Reward penalty coefficient $\lambda$ | 0.1 |
| **BOSA** | |
| Temperature coefficient | 0.2 |
| Maximum log std | 2 |
| Minimum log std | $-20$ |
| Policy regularization coefficient $\lambda_{\text{policy}}$ | 0.1 |
| Transition coefficient $\lambda_{\text{transition}}$ | 0.1 |
| Threshold parameter $\epsilon, \epsilon'$ | $\log(0.01)$ |
| Value weight $\omega$ | 0.1 |
| CVAE ensemble size of the dynamics model | 5 |
| **SRPO** | |
| Discriminator network | (256, 256) |
| Data selection ratio | 0.5 |
| Reward coefficient $\lambda$ | $\{0.1, 0.3\}$ |
| **IGDF** | |
| Representation dimension | $\{16, 64\}$ |
| Contrastive encoder network | (256, 256) |
| Encoder pretraining steps | 7000 |
| Importance coefficient | 1.0 |
| Data selection ratio $\xi\%$ | 75% |
| **OTDF** | |
| CVAE training steps | 10000 |
| CVAE learning rate | 0.001 |
| Number of sampled latent variables $M$ | 10 |
| Standard deviation of Gaussian distribution | $\sqrt{0.1}$ |
| Cost function | cosine |
| Data selection ratio $\xi\%$ | 80% |
| Policy coefficient $\beta$ | $\{0.1, 0.5\}$ |
| **STC** | |
| Correction threshold $\lambda$ | $\{1.0, 5.0\}$ |
| Reward gradient coefficient $\alpha$ | 0.5 |
| Q-weighted loss coefficient $\beta$ | $\{0.5, 5.0\}$ |

Table 9: **Performance comparison under friction shift**. med = medium, e = expert. The **Source** column means the source domain dataset, and the **Target** column indicates the target domain dataset quality. The normalized average scores in the target domain across 5 seeds are reported and $\pm$ captures the standard deviation. We highlight the best cell.

| Source | Target | IQL | DARA | BOSA | SRPO | IGDF | OTDF | STC (ours) |
|---|---|---|---|---|---|---|---|---|
| halfcheetah-med | med | **69.7**±1.3 | 66.2±4.1 | 68.5±1.2 | 66.7±2.1 | 66.1±1.9 | 66.8±0.6 | 67.4±1.0 |
| halfcheetah-med | med-e | 66.8±0.6 | 60.4±11.9 | 69.2±0.6 | 66.8±3.9 | 68.6±0.6 | 61.3±5.4 | **69.4**±1.0 |
| halfcheetah-med | expert | 68.2±0.4 | 67.9±0.5 | **69.7**±1.3 | 67.1±2.3 | 68.2±2.5 | 68.3±0.6 | 67.3±0.6 |
| hopper-med | med | 24.5±1.7 | 24.2±2.2 | 25.4±1.9 | 26.5±2.0 | 27.2±4.3 | **29.3**±4.6 | 27.4±2.0 |
| hopper-med | med-e | **26.4**±2.5 | 21.7±5.6 | 23.0±1.7 | 24.3±3.7 | 25.7±2.3 | 26.4±2.4 | 23.4±5.1 |
| hopper-med | expert | 21.5±1.1 | 24.6±2.5 | 25.7±0.8 | 21.1±2.7 | 23.6±3.8 | 25.9±1.1 | **41.6**±26.1 |
| walker2d-med | med | 72.6±6.2 | **73.9**±11.7 | 72.1±5.2 | 67.5±6.0 | 70.3±3.6 | 70.4±6.6 | 73.6±8.6 |
| walker2d-med | med-e | 59.0±1.3 | 61.5±20.2 | 42.8±17.5 | 57.4±7.9 | 59.1±5.4 | 59.9±10.2 | **71.2**±5.8 |
| walker2d-med | expert | 52.5±3.6 | 60.0±11.7 | 51.5±12.9 | 52.3±12.7 | 54.9±6.1 | 49.6±14.6 | **67.9**±6.6 |
| ant-med | med | 58.2±2.3 | 58.7±2.0 | 57.6±4.0 | 56.9±2.5 | 55.2±3.2 | 58.3±0.2 | **60.2**±3.1 |
| ant-med | med-e | 59.3±2.5 | 58.3±1.0 | **60.5**±0.3 | 58.4±0.4 | 57.8±0.5 | 58.1±0.4 | 45.7±10.5 |
| ant-med | expert | 58.2±0.3 | 58.3±0.5 | 59.3±1.8 | 57.2±2.0 | 58.3±0.2 | 56.5±2.3 | **60.6**±1.8 |
| Total Score | | 636.8 | 635.7 | 625.2 | 622.2 | 635.1 | 630.8 | **675.6** |

# E   Additional Experimental Results

## E.1   Results Under Friction Shift

We summarize the normalized score comparison of STC against other baselines under the friction shift tasks in Table 9. STC achieves the best overall performance across 12 tasks. We observe that in the friction shift task, the performance gap between different algorithms is relatively small, possibly due to the minor discrepancy between the source and target domains under this type of shift. Nevertheless, our method still outperforms all baselines in terms of the total score.

## E.2   Additional Visualization Results for STC Correction

This section provides additional visualizations illustrating how STC improves the alignment between transition distributions in the source and target domains. Specifically, we include further visualizations on ant environment with gravity shift in Figure 5. We first apply STC to the original source transitions to obtain corrected transitions. For each target transition, we find the nearest neighbor in the original source dataset based on the state pair $(s, s')$, and extract the corresponding action $a_{\text{src}}$. The matching corrected action $\hat{a}_{\text{src}}$ is then retrieved from the STC-processed dataset. We plot kernel density estimation (KDE) curves for both $a_{\text{src}}$ and $\hat{a}_{\text{src}}$, and compare them with the target domain action distribution. As shown in Figure 5, we observe that the corrected distribution (green curves) aligns more closely with the target distribution (blue

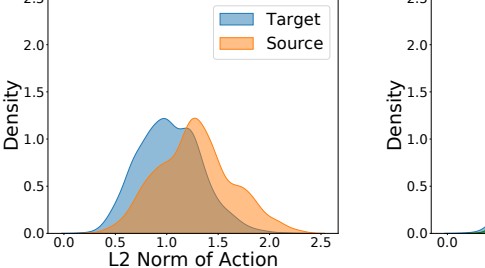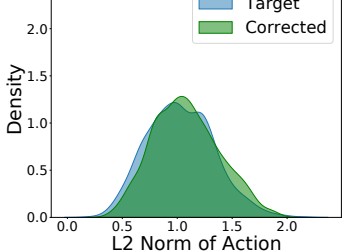

Figure 5: **Action distribution comparison on ant environment with gravity shift.** The left panel shows KDE curves comparing source domain actions and target domain actions, the right panel shows KDE curves comparing STC-corrected source actions with target actions.

Table 10: Percentage of source-domain transitions corrected by the selective mechanism across different tasks.

| Environment | Correction Percentage |
|---|---|
| hopper-gravity-shift | 89.84% |
| ant-gravity-shift | 96.88% |
| walker2d-gravity-shift | 25.78% |
| halfcheetah-gravity-shift | 31.25% |
| hopper-morphology-shift | 91.41% |
| ant-morphology-shift | 92.19% |
| walker2d-morphology-shift | 93.75% |
| halfcheetah-morphology-shift | 42.19% |

curves) than the original one (orange curves), demonstrating STC's effectiveness in reducing the distribution gap.

### E.3   Correction Rate of Source-Domain Transitions

After applying the selective correction mechanism, the proportion of corrected source-domain transitions varies across tasks, depending on the selection threshold $\lambda$ and the reliability of inverse policy predictions evaluated by the forward dynamics model. Specifically, a source transition is modified only when the corrected action leads to a predicted next state that is closer to the target dynamics than that induced by the original action. As a result, the correction rate is inherently task-dependent and reflects the confidence of the learned models. To quantify the extent of modification, we report the percentage of corrected source-domain transitions across representative tasks in Table 10. The results indicate a substantial correction ratio in most tasks, highlighting the importance of source transition correction for the effectiveness of our method.

### E.4   Ablation Study on Q-weighted Loss Coefficient

The coefficient $\beta$ balances Q-value maximization and behavior cloning. We evaluate $\beta \in \{0.5, 5.0, 10.0\}$, as shown in Figure 6. Some environments are sensitive to $\beta$, while others are not. We use $\beta = 0.5$ or $5.0$ across all tasks for good overall performance, with $5.0$ being the most common choice.

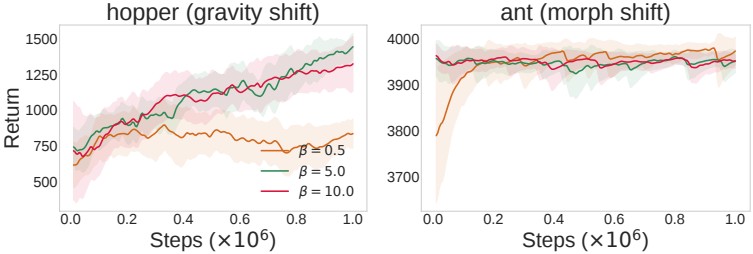

Figure 6: Q-weighted loss coefficient $\beta$. We report target domain return results in two shift tasks with different $\beta$. The shaded region captures the standard deviation.

### E.5   Extension to Online Learning Settings

Though our method is designed for the offline cross-domain setting, the core idea of Selective Transition Correction is transferable to online scenarios where both the source and target domains are online. To evaluate our method in online settings, we develop an online variant of STC using SAC Haarnoja et al. (2018) as the base algorithm. At each training step, we additionally collect an additional mini-batch of fresh interactions from the target environment and use this data to update our dynamics and reward models

Table 11: Performance comparison on online settings.The normalized average scores in the target domain across 5 seeds are reported and ± captures the standard deviation. We highlight the best cell.

| Type | Task | SAC | DARC | PAR | VGDF | STC |
|------|------|-----|------|-----|------|-----|
| Gravity | halfcheetah-gravity | 24.57±16.64 | **62.54**±3.78 | 59.09±3.97 | 43.16±25.53 | 57.45±5.47 |
| | walker2d-gravity | **68.73**±6.50 | 6.19±0.70 | 67.35±9.97 | 62.00±5.08 | 65.58±6.07 |
| | ant-gravity | **85.85**±1.50 | 40.63±4.45 | 82.85±4.49 | 51.79±20.77 | 83.36±5.12 |
| | hopper-gravity | 96.79±2.69 | 11.16±1.70 | 91.52±17.01 | 90.83±17.11 | **98.44**±13.96 |
| Kinematic | halfcheetah-footjnt-medium | 33.14±26.10 | 33.65±27.41 | **88.09**±3.13 | 39.67±17.70 | 58.37±23.43 |
| | halfcheetah-footjnt-hard | 46.82±22.95 | 37.90±24.50 | 54.51±31.71 | 24.77±11.49 | **61.40**±24.30 |
| | walker2d-footjnt-medium | 67.10±22.61 | 90.71±10.97 | 72.16±15.95 | 38.44±23.49 | **81.85**±6.50 |
| | walker2d-footjnt-hard | 41.33±13.83 | 13.32±1.48 | 37.75±15.40 | 45.27±15.82 | **51.13**±11.66 |
| | ant-anklejnt-medium | 110.10±3.89 | 84.36±6.14 | 115.00±5.60 | 55.34±11.97 | **115.51**±4.43 |
| | ant-anklejnt-hard | **117.14**±2.85 | 66.54±21.62 | 114.33±2.94 | 59.58±16.91 | 108.40±3.39 |
| | hopper-footjnt-medium | 84.38±30.13 | 14.47±0.43 | 98.28±2.08 | **105.72**±2.65 | 101.48±3.78 |
| | hopper-footjnt-hard | 36.83±9.14 | 26.54±11.07 | 29.56±3.75 | **47.34**±23.57 | 29.62±6.27 |
| | Total Score | 812.81 | 488.01 | 910.47 | 663.90 | **912.58** |

Table 12: **Comparison between target-only baseline and STC.** The normalized average scores in the target domain across 5 seeds are reported and ± captures the standard deviation.

| Type | Source | Target | Target-only | STC (ours) |
|------|--------|--------|-------------|------------|
| Gravity | halfCheetah-medium | medium | 0.6±0.5 | **42.4±5.3** |
| | halfCheetah-medium | expert | 0.0±0.1 | **37.3±1.5** |
| | hopper-medium | medium | 46.2±3.7 | 43.4±6.1 |
| | hopper-medium | expert | 18.6±13.1 | 23.4±6.4 |
| | walker2d-medium | medium | 28.5±7.4 | **41.6±4.0** |
| | walker2d-medium | expert | 17.9±11.4 | **36.9±6.5** |
| | ant-medium | medium | 32.0±7.6 | 42.6±8.4 |
| | ant-medium | expert | 16.9±4.5 | 23.2±6.1 |
| Morph | halfCheetah-medium | medium | 12.3±3.7 | 19.5±2.2 |
| | halfCheetah-medium | expert | -0.7±0.6 | 7.7±1.2 |
| | hopper-medium | medium | 42.5±17.3 | **43.1±23.9** |
| | hopper-medium | expert | 62.3±28.0 | **53.2±50.3** |
| | walker2d-medium | medium | 43.1±11.5 | **56.7±8.1** |
| | walker2d-medium | expert | 9.9±4.1 | 32.9±7.3 |
| | ant-medium | medium | 76.8±2.8 | **77.2±2.8** |
| | ant-medium | expert | 50.3±12.2 | **74.1±21.2** |
| | Total Score | | 457.2 | **655.2** |

in real-time. The updated models are applied to correct the source domain data for the current iteration. We compare our method against several state-of-the-art online off-dynamics RL algorithms, including SAC Haarnoja et al. (2018), DARC Eysenbach et al. (2021), PAR Lyu et al. (2024a) and VGDF Xu et al. (2023), across multiple challenging environments from the ODRL Lyu et al. (2024b) benchmark. We report the performance of the methods on representative gravity-shift and kinematic-shift tasks in Table 11. The results show that STC maintains strong performance on online setting, achieving significant improvements over baseline methods across numerous tasks.

## E.6 Comparison with Target-Only Baseline

To isolate the contribution of corrected source-domain transitions, we compare STC with a target-only baseline that uses the same offline RL backbone but is trained exclusively on the target-domain dataset without incorporating any corrected source data. As shown in Table 12, the target-only baseline consistently underperforms STC across most tasks, with particularly large gaps in *halfcheetah* and *walker2d* environments. In several cases, training solely on the limited target dataset leads to unstable learning or severely degraded performance. Overall, STC achieves a substantially higher total score (655.2 vs. 457.2), demonstrating that incorporating corrected source transitions provides consistent improvements over target-only learning.

# F   Clarifications on Algorithm Design

In this section, we provide additional clarifications on two key design choices in our algorithm and discuss the motivation behind them.

**Introducing forward dynamics model for selective correction mechanism.**  Theoretically, a perfectly accurate inverse policy would be sufficient to correct source transitions and reduce dynamics mismatch. In practice, however, the limited target dataset (5,000 transitions) makes it difficult for the inverse policy model to achieve ideal performance, and inaccurate action corrections can introduce additional errors. To address this issue, rather than fully trusting all corrections produced by the inverse policy model, we introduce a forward dynamics model to enforce a more conservative selective correction mechanism. Specifically, the forward model evaluates the reliability of corrected transitions by comparing prediction errors and selectively accepts or rejects corrections accordingly, ensuring that only dynamically consistent corrections are applied. By mutually constraining each other, the inverse and forward models help reduce the impact of errors from any single model. This selective correction mechanism stabilizes the correction process and yields more reliable data for offline learning.

**Using first-order Taylor approximation for reward correction.**  We use a first-order Taylor expansion to approximate the reward of corrected actions instead of directly predicting rewards with a reward model. Direct reward prediction in off-dynamics offline RL is sensitive to model bias and distribution shift, which is more likely to cause instability and uncontrollable error accumulation. We directly use the inverse policy model for action correction because we can assess its reliability through the forward dynamics model via our selective correction mechanism. In contrast, we lack a comparable and reliable way to evaluate the bias of predicted rewards. The Taylor-based approximation provides a stable and conservative reward adjustment by leveraging local reward gradients around well-supported source actions. We further normalize and clip the gradients and introduce a scaling hyperparameter $\alpha$ to control the magnitude of reward correction, resulting in a bias-controlled and robust design for cross-domain offline RL.

**Discussion on identifiability of the inverse correction step.**  In general, inverse dynamics in reinforcement learning environments are not strictly identifiable, as multiple actions may induce similar state transitions. Our method does not rely on exact identifiability of the inverse mapping. Instead, the inverse policy model is trained to generate actions consistent with the target-domain dynamics, rather than recover a unique ground-truth action. Since it is trained on target-domain data generated by a specific behavior policy, it captures representative action patterns induced by this policy and models a plausible mode of the conditional action distribution given observed transitions, rather than all valid actions. This ambiguity is further mitigated by the selective correction mechanism, where corrected actions are evaluated using a learned forward dynamics model and only adopted when they lead to transitions better aligned with the target dynamics, thereby reducing the requirement for exact inverse identifiability and restricting correction to actions consistent with the target environment under the learned dynamics. Empirically, the distribution of corrected actions closely matches that of the target domain, suggesting that the combination of inverse modeling and selective correction yields meaningful and stable corrections in practice.

**Robustness to imperfect learned components.**  STC relies on three learned components, namely an inverse policy model, a forward dynamics model, and a reward model, while remaining robust to moderate inaccuracies in any individual component. These components serve complementary roles: the inverse model proposes candidate action corrections, the forward model evaluates their consistency with the target dynamics, and the reward model is only used for a first-order reward adjustment. Consequently, the method does not form a strictly sequential pipeline in which errors are directly accumulated. The selective correction mechanism further enforces a conservative update rule, where a corrected action is adopted only if it yields improved alignment with the target dynamics under the forward model, thereby introducing a form of mutual consistency check that mitigates the effect of inaccurate predictions. In addition, the restricted use of the reward model limits the propagation of estimation errors. Empirically, even with only 5k target-domain transitions, the learned components provide effective correction signals.

## G  Compute Infrastructure

We list the compute infrastructure that we use to run all algorithms adopted in this paper in Table 13.

Table 13: Computing infrastructure used to run all algorithms evaluated in this paper.

| Component | Specification |
|-----------|---------------|
| CPU | AMD EPYC 7452 |
| GPU | RTX3090$\times$8 |
| Memory | 288GB |

