# OpenReview forum: "Cross-Domain Offline Policy Adaptation via Selective Transition Correction"
_TMLR — Under review for TMLR_

### Review · Reviewer_bq7v · 2026-04-16

**Summary Of Contributions:**

This paper studies cross-domain offline reinforcement learning with a large source-domain dataset and a limited target-domain dataset under dynamics mismatch. The main idea is to *correct* source transitions rather than merely filter or penalize them. Concretely, the method trains an inverse model to infer a target-consistent action from $(s,s')$, a reward model to adjust the reward after action correction, and a forward dynamics model to decide whether the corrected transition should be retained. The resulting method, STC, is evaluated on ODRL-style MuJoCo benchmarks under gravity, friction, and morphology shifts, with additional studies on target-data quality, target-data size, and action-distribution alignment.

**Audience:**

Yes

**Audience Explanation:**

Cross-domain offline RL under dynamics mismatch is a relevant and timely problem.

**Claims And Evidence:**

No

**Claims Explanation:**

On the positive side, the empirical section is broad and, at large, convincing. The method is evaluated across several tasks, several kinds of dynamics shift, multiple source/target quality combinations, and different target dataset sizes. The aggregate results are favorable to STC, and the visualization section also supports the claim that corrected actions become better aligned with the target-domain action distribution. The paper is also well written: the method's intended mechanism is easy to understand.

That said, I do not think the theoretical part currently provides solid support for the paper's stronger claims.

First, Assumption 4.2 appears to absorb much of the real difficulty of the problem. The central challenge is precisely whether an inverse-model-based correction can be learned reliably from limited target data under domain shift. Yet the theory assumes that the estimated inverse policy is already close enough to the target behavior policy, and the main bound then follows by combining that approximation term with the source-target dynamics discrepancy. This is not wrong as a formal statement, but it leaves open the essential question of when the inverse correction step is actually justified. Naturally, this might be a common assumption in the literature (which I am not an expert on), but I thought I should raise the point.

Second, I am not convinced by Theorem 4.5 in the continuous control setting used in the experiments. As written, the argument seems to rely on turning a bound involving action differences into a bound involving total variation distance between behavior policies. In continuous action spaces, this step is not immediate and, in the present form, appears unjustified. Since the experiments are precisely in continuous control MuJoCo domains, this is not a minor issue.

Third, Theorem 4.6 is stated as a finite data bound involving cardinalities such as $|S|$ and $|S \times A \times S|$. This strongly suggests a finite/tabular viewpoint, whereas the empirical setting is continuous and function-approximation-based. I do not object to including a simplified theorem for intuition, but then the scope and limitations of the guarantee must be stated much more explicitly.

**Requested Changes:**

Under what conditions is the inverse correction step identifiable? If multiple actions can induce similar transitions, why should the inverse model recover a meaningful target-consistent action?

What is the empirical contribution of the selective mechanism by itself? In particular, what happens with inverse correction but *without* forward-model-based selection?

How important is the reward correction term? A dedicated ablation comparing action correction only vs. action + reward correction would make the contribution decomposition much clearer.

What fraction of source transitions are actually corrected and retained in each environment? This would help understand whether the method behaves more like conservative filtering or genuine large-scale correction.

**Important Changes**

Repair or substantially reframe the theory section, i.e. Theorem 4.5 and the scope of Theorem 4.6.

Add a direct ablation isolating the role of selective correction, i.e. correction with and without the forward-model acceptance test.

**Suggested Changes that would Improve the Paper**

Add more explicit discussion of failure cases and tasks where STC is not the best method.

Report correction/acceptance ratios and related diagnostics for each domain.

Include more targeted ablations for the reward correction mechanism.

Clarify the computational overhead in the main paper rather than leaving it mostly to the appendix.

---

> ### Author Response · Authors · 2026-05-02
> **Author Responses to Reviewer bq7v (part 1)**
>
> We thank the reviewer for careful reading of our paper and the valuable comments. We address the reviewer’s concerns point by point, and we hope that our clarifications can help resolve them. We have also revised the manuscript accordingly based on the reviewer’s suggestions to further improve its quality.
>
> **Point-by-Point Responses to the Reviewer’s Concerns**
>
> **Concern 1: Assumption 4.2 appears to absorb much of the real difficulty of the problem.**
>
> We appreciate the reviewer's comment regarding Assumption 4.2. We agree that the reliability of inverse-model-based correction under limited target data is a central challenge, but Assumption 4.2 is not meant to bypass the core difficulty. We would like to clarify the role of this assumption.
>
> Assumption 4.2 should be viewed as an idealized condition introduced for theoretical analysis, rather than as a practical claim that the inverse policy can always be learned with high accuracy. Its purpose is to establish a feasibility foundation for the proposed approach: under such idealized conditions, correcting source transitions via an inverse dynamics model can effectively mitigate dynamics mismatch and lead to improved performance. This assumption is clearly satisfied when the target domain provides sufficiently abundant data. In practice, the inverse policy model may indeed be biased due to limited data, which is precisely the issue that the forward dynamics model addresses through selective correction.
>
> Empirically, as shown in Figure 2 of the submitted manuscript, the distribution of corrected actions closely resembles that of the target domain, suggesting that the learned inverse policy together with the selective correction mechanism produces meaningful corrections even under limited target data, which provides empirical support for the assumption.
>
>
>
> **Concern 2: Theorem 4.5 in the continuous control setting used in the experiments may not be convincing.**
>
>
> We thank the reviewer for the insightful concern. We agree that directly relating action differences to total variation (TV) distance is not immediate in continuous action spaces. We would like to clarify that Theorem 4.5 (as well as Theorems 4.4 and 4.6) is derived under a tabular (finite, discrete) setting, where policies induce discrete action distributions. In this setting, the total variation distance naturally arises as a measure of discrepancy between behavior policies (i.e., $\mu\_{\mathrm{src}} \text{ and } \mu_{\mathrm{tar}}$), and the bound characterizes how such distributional differences propagate to value estimation.
>
> Extending this analysis to continuous action spaces with function approximation would require substantially more involved technical tools. Instead, we adopt a simplified tabular formulation to obtain interpretable insights into how policy mismatch affects value estimation. In particular, the result highlights that the Q-value deviation is controlled by the divergence between behavior policies, which provides theoretical support for aligning source and target action distributions in our method.
>
> This type of simplified analysis is also adopted in some cross-domain RL papers (e.g., OTDF [1], IGDF [2]), where results are derived in discrete settings and used to guide deep RL algorithm design. We will revise the manuscript to explicitly clarify the tabular assumption and the scope of the guarantee.
>
>
> **Concern 3: The scope and limitations of the guarantee for Theorem 4.6 must be stated much more explicitly.**
>
>
> We thank the reviewer for this important observation. We agree that Theorem 4.6 is derived under a finite (tabular) MDP setting, as reflected by the appearance of cardinalities such as $|\mathcal{S}|$ and $|\mathcal{S} \times \mathcal{A} \times \mathcal{S}|$, which arise from standard finite-sample analysis. Our intention is not to claim that this bound directly characterizes the continuous control setting with function approximation used in experiments. Instead, the theorem provides a simplified analysis that highlights the key factors governing cross-domain performance, in particular the role of the inverse policy error $\kappa$. Specifically, it shows that the performance gap is controlled by these quantities, explaining why accurate inverse modeling are critical for effective transition correction.
>
> Such use of simplified tabular analysis to extract structural insights is common in offline reinforcement learning (e.g., OTDF [1], IGDF [2]), where theoretical results are established in finite settings and used to guide deep RL algorithm design (akin to our response to Concern 2 above).
>
> We will revise the manuscript to explicitly clarify the scope and limitations of Theorem 4.6, emphasizing that it is intended as an interpretable abstraction rather than a tight guarantee for continuous control.

---

> > ### Author Response · Authors · 2026-05-02
> > **Author Responses to Reviewer bq7v (part 2)**
> >
> > **Concern 4: Under what conditions is the inverse correction step identifiable?**
> >
> >
> > Thanks for the comment. In general, the environment inverse dynamics may be non-identifiable due to the existence of multiple actions leading to similar transitions. Our method does not require exact identifiability of the inverse mapping. Instead, the goal of the inverse model is to produce an action that is consistent with the target-domain dynamics, rather than to recover the unique ground-truth action. Since the inverse model is trained on target-domain data generated by a specific behavior policy, it learns to capture representative action patterns induced by this policy. In other words, it models a plausible mode of the conditional action distribution given observed transitions, rather than all possible valid actions.
> >
> > Importantly, the ambiguity arising from non-identifiability is further mitigated by our selective correction mechanism. Rather than directly relying on the inverse prediction, we evaluate corrected actions using the forward dynamics model and only adopt them when they lead to transitions that better match the target dynamics. This effectively reduces the problem from exact inverse recovery to selecting actions that are most consistent with the target environment.
> >
> > Empirically, as shown in Figure~2, the corrected action distribution aligns closely with that of the target domain, indicating that the combination of inverse modeling and selective correction produces meaningful and stable corrections in practice.
> >
> > We have added a discussion on the identifiability of the inverse correction step in Appendix F of the revised manuscript. The corresponding revisions are highlighted in red.
> >
> > **Concern 5: What is the empirical contribution of the selective mechanism by itself?**
> >
> > We thank the reviewer for this question. To evaluate the contribution of the selective mechanism, we conduct an ablation study comparing our full method (STC) with a variant that applies inverse correction without forward-model-based selection. We present the comparison in the following Table 1.
> >
> > As shown in the results, removing the selective mechanism generally leads to noticeable performance degradation across several tasks (e.g., halfcheetah-gravity, halfcheetah-morph, hopper-gravity), where the full method consistently achieves higher returns. This suggests that directly applying inverse corrections may introduce noisy or misaligned actions, and the selective mechanism helps filter out such harmful corrections. In a few cases (e.g., walker2d-morph), the gap is smaller or slightly reversed, suggesting that when inverse correction is already sufficiently accurate, the additional benefit of selection may be more limited. Nevertheless, the overall trend is clear: the selective mechanism is an important component for achieving reliable gains from source transition correction.
> >
> > | Type    | Source                     | Target | w/o selection | STC (ours) |
> > |---------|----------------------------|--------|----------------|-------------|
> > | Gravity | walker2d-medium            | medium | 39.0±4.1    | **41.6±4.0** |
> > | Gravity | halfcheetah-medium         | medium | 27.6±5.6     | **42.4±5.3** |
> > | Gravity | hopper-medium              | medium | 37.9±5.5     | **43.4±6.1** |
> > | Morph   | halfcheetah-medium   | medium | 4.2±8.4      | **19.5±2.2** |
> > | Morph   | walker2d-medium        | medium | **59.7±8.3** | 56.7±8.1 |
> > | Morph   | hopper-medium         | medium | 42.8±7.7 | **43.1±23.9** |
> > | **Total Score**     |                            |      | 211.2         | **246.80** |
> >
> > Table 1: Ablation study on the effect of the selective mechanism for source transition correction.

---

> > > ### Author Response · Authors · 2026-05-02
> > > **Author Responses to Reviewer bq7v (part 3)**
> > >
> > > **Concern 6: How important is the reward correction term? A dedicated ablation comparing action correction only vs. action + reward correction would make the contribution decomposition much clearer.**
> > >
> > > We thank the reviewer for this suggestion. The reward correction term does provide a consistent positive contribution to performance. In Section 5.2 (Parameter Study) of the submitted manuscript, we analyze the effect of the reward correction coefficient $\alpha$. Notably, setting $\alpha = 0$ corresponds to removing reward correction entirely. As shown in Figure 3(b), properly choosing $\alpha$ consistently improves performance over the action correction only case across most tasks.
> > >
> > > To further clarify the contribution decomposition, we conduct an additional ablation study comparing STC with a variant that removes reward correction (i.e., action correction only) across six tasks. As shown in the following Table 2, action correction already accounts for the majority of the performance gain, while reward correction provides a consistent additional improvement under several challenging domain shifts, such as *halfcheetah-morph-shift* and *hopper-gravity-shift*. Overall, STC achieves a higher total return, indicating that reward correction contributes improvement rather than being the primary driver.
> > >
> > > These results suggest that reward correction plays a stabilizing role by compensating the bias introduced by action modification, leading to better performance across tasks.
> > >
> > >
> > > | Type    | Source                                | Target |Action Only | STC (ours) |
> > > |---------|---------------------------------------|--------|------------------------|-------------|
> > > | Gravity | walker2d-medium               | medium | 35.9 ±2.8             | **41.6±4.0** |
> > > | Gravity | halfcheetah-medium            | medium | 38.1± 1.0             | **42.4±5.3** |
> > > | Gravity | hopper-medium                 | medium | 40.5±5.9             | **43.4±6.1** |
> > > | Morph   | walker2d-medium             | medium | 56.4±9.9         | 56.7±8.1 |
> > > | Morph   | halfcheetah-medium        | medium | 17.4±5.9            | **19.5±2.2** |
> > > | Morph   | hopper-medium              | medium | 33.6±12.1            | **43.1±23.9** |
> > >
> > > Table 2: Ablation study comparing action correction only vs. action + reward correction.
> > >
> > > **Concern 7: What fraction of source transitions are actually corrected and retained in each environment?**
> > >
> > > We thank the reviewer for raising this point. After the selective correction process, the number of modified transitions varies across tasks depending on the correction threshold $\lambda$ and the reliability of the inverse policy predictions as evaluated by the forward dynamics model. Typically, only a subset of source transitions that satisfy the selection criterion (i.e., the predicted next state under the corrected action is closer to the target dynamics than the original) are adopted.
> > > We report below the percentage of source-domain transitions corrected for several tasks to provide a rough indication of the extent of modifications. The results in following Table 3 indicate a substantial correction ratio in most tasks, highlighting the importance of source transition correction for the effectiveness of our method.
> > >
> > > | Environment                   | Correction percentage |
> > > |------------------------------|------------------------|
> > > | hopper-gravity-shift         | 89.84\%                |
> > > | ant-gravity-shift            | 96.88\%                |
> > > | walker2d-gravity-shift       | 25.78\%                |
> > > | halfcheetah-gravity-shift    | 31.25\%                |
> > > | hopper-morphology-shift      | 91.41\%                |
> > > | ant-morphology-shift         | 92.19\%                |
> > > | walker2d-morphology-shift    | 93.75\%                |
> > > | halfcheetah-morphology-shift | 42.19\%                |
> > >
> > > Table 3. The percentage of source-domain transitions corrected.
> > >
> > > **Changes Made in the Revised Manuscript**
> > >
> > > Based on the reviewer’s suggestions, we have revised the manuscript accordingly. The changes are detailed as follows:
> > >
> > > **Change 1: Repair or substantially reframe the theory section, i.e. Theorem 4.5 and the scope of Theorem 4.6.**
> > >
> > > We have revised the Theoretical Analysis section to clarify the assumptions and improve the rigor of the analysis. In particular, we clarify that our theoretical setting is the tabular case, and we further discuss the gap between the tabular results and the experimental setting, as well as how the theory provides guidance for the empirical design. The corresponding revisions are highlighted in red.

---

> > > > ### Author Response · Authors · 2026-05-02
> > > > **Author Responses to Reviewer bq7v (part 3)**
> > > >
> > > > **Change 2: Add a direct ablation isolating the role of selective correction**
> > > >
> > > > We include a dedicated ablation study to isolate the effect of the selective correction mechanism. Specifically, we report performance comparisons across multiple tasks with and without the selective mechanism, which provides a clearer understanding of its contribution. The corresponding revisions have been incorporated into Section 5.5 of the revised manuscript and are highlighted in red.
> > > >
> > > >
> > > >
> > > > **Change 3:Add more explicit discussion of failure cases and tasks where STC is not the best method.**
> > > >
> > > > We thank the reviewer for this suggestion. A more explicit discussion of failure cases can help clarify the scope and limitations of STC.
> > > >
> > > > From the experimental results, STC does not achieve the best performance in every setting, such as the *walker2d-gravity-shift* environment with the medium-expert source dataset and the *halfcheetah-morphology-shift* environment with the limited target dataset. This suggests that the gains from transition correction may be less pronounced on a small subset of tasks. In particular, when the domain shift is relatively mild or the source transitions are already well aligned with the target dynamics, the additional benefit of correction can be limited. More generally, STC tends to be more effective when correction improves cross-domain alignment without introducing excessive perturbation.
> > > >
> > > > We also note that our hyperparameter selection is intentionally kept largely consistent across tasks to ensure a fair and comparable evaluation. Specifically, we tune $\lambda$ from \{1.0, 5.0\} for all tasks and use a fixed $\alpha$ across environments. While this protocol better reflects the overall robustness of the method and yields strong average performance, it may not be optimal for every individual task. As a result, the relatively weaker performance of STC on a few tasks also reflects our choice to balance strong overall performance with consistent hyperparameter settings across environments, rather than optimizing separately for each individual task.
> > > >
> > > > We include a discussion of these cases in Section 5.1 （Main Results) of the revised paper. We expand the discussion to include representative failure cases,  The corresponding revisions are highlighted in red.
> > > >
> > > >
> > > > **Change 4: Report correction/acceptance ratios and related diagnostics for each domain.**
> > > >
> > > > We report correction and acceptance ratios across domains along with additional diagnostics in the revised paper, to provide better insight into when and how the method modifies source transitions. The corresponding revisions have been incorporated into Appendix E.3 of the revised manuscript and are highlighted in red.
> > > >
> > > > **Change 5:Include more targeted ablations for the reward correction mechanism.**
> > > >
> > > > We add targeted ablation studies to better disentangle the role of reward correction and analyze its impact on performance and stability. The corresponding revisions have been incorporated into Section 5.5 of the revised manuscript and are highlighted in red.
> > > >
> > > >
> > > >
> > > > **Change 6:Clarify the computational overhead in the main paper rather than leaving it mostly to the appendix.**
> > > >
> > > > We move the discussion of computational overhead to Section 6 of the main paper. The corresponding revisions are highlighted in red.
> > > >
> > > >
> > > > **Change 7: Discussion on the identifiability of the inverse correction step.**
> > > >
> > > > We have added a discussion on the identifiability of the inverse correction step in Appendix F of the revised manuscript. The corresponding revisions are highlighted in red.
> > > >
> > > >
> > > >  We hope these can resolve the concerns. If there are any further questions or concerns, please feel free to let us know.
> > > >
> > > > [1] Cross-domain offline policy adaptation with optimal transport and dataset constraint.
> > > >
> > > > [2] Contrastive representation for data filtering in cross-domain offline reinforcement learning.

---

> > > > > ### Comment · Reviewer_bq7v · 2026-05-02
> > > > >
> > > > > I thank the authors for their detailed and constructive response. I have read the response and the described revisions carefully. Overall, I think the authors have addressed a substantial fraction of my concerns, especially on the empirical and presentation sides.
> > > > >
> > > > > The main point that remains is theoretical scope: the authors do not provide a new guarantee for the continuous-control setting used in the experiments. Instead, they clarify that Theorems 4.4-4.6 should be understood in a finite/tabular setting and are intended as an interpretable abstraction rather than as a tight guarantee for MuJoCo-style continuous control with function approximation. I think this is a reasonable reframing, provided that **the revised manuscript states this limitation clearly.**
> > > > >
> > > > > The authors also responded well to the empirical concerns. I appreciate the new ablation isolating the selective correction mechanism. The reported comparison between full STC and correction without forward-model-based selection supports the claim that the selective mechanism is useful overall: the full method obtains a higher total score, even though it is not uniformly better on every single task. This is exactly the kind of diagnostic that was needed to support the method's design.
> > > > >
> > > > > Similarly, the additional action-only versus action-plus-reward-correction ablation is useful. The results suggest that action correction accounts for much of the gain, while reward correction provides a consistent additional improvement and some stabilization. This makes the contribution decomposition clearer.
> > > > >
> > > > > I also appreciate the correction percentage diagnostics. The fact that the percentage of corrected transitions varies substantially across tasks is informative. It helps clarify that STC is not merely applying a uniform relabeling operation, but is adapting the amount of correction depending on the learned forward/inverse consistency. The added discussion of failure cases is also helpful. In particular, it is useful that the authors acknowledge regimes where the benefit of correction is smaller, for example when the domain shift is mild or the original source transitions are already well aligned with the target dynamics.
> > > > >
> > > > > The response on identifiability is also satisfactory at a practical level. The authors clarify that exact inverse dynamics identifiability is not required; rather, the inverse model is used to produce a plausible target-consistent action, and the forward model selection step acts as a safeguard against bad or ambiguous corrections. This is not a formal identifiability result, but it is a reasonable description of the algorithmic mechanism.
> > > > >
> > > > > Overall, I now view the paper more positively. The empirical contribution is meaningful, the method is simple and interesting, and the added ablations materially strengthen the evidence for the design choices. My remaining reservation is that the theory should be framed carefully as a simplified analysis rather than as a guarantee for the continuous control experiments. Assuming the revised manuscript makes this limitation explicit, I think the paper satisfies the TMLR acceptance criteria.

---

> > > > > > ### Author Response · Authors · 2026-05-02
> > > > > > **Author Responses to Reviewer bq7v**
> > > > > >
> > > > > > We thank the reviewer for the careful reading of our response and for the positive and helpful feedback.
> > > > > >
> > > > > > Regarding the remaining concern on the theoretical scope, we agree that the theory should be framed carefully as a simplified analysis rather than as a guarantee for the continuous-control experiments. In the revised manuscript, we have explicitly clarified this point in Section 4.2 (Theoretical Analysis). In particular, we now clearly state that the analysis is conducted under a tabular (finite state-action) MDP setting and is intended to provide qualitative insight into the role of source transition correction, rather than a tight or formal guarantee for continuous-control problems.
> > > > > >
> > > > > > We further emphasize that the purpose of the theoretical analysis is to offer intuition and partial justification for the proposed design under idealized conditions, and to help explain how transition correction can improve cross-domain alignment, rather than to fully characterize the behavior of the method in practical continuous-control scenarios. Extending the analysis to continuous-control settings can be a direction for future work.
> > > > > >
> > > > > > We hope that this revision addresses the reviewer’s concern, and we appreciate the suggestion, which helps us improve the clarity and positioning of the theoretical contribution.

---

### Review · Reviewer_pxSH · 2026-04-20

**Summary Of Contributions:**

This paper tackles cross-domain offline reinforcement learning, where a key challenge is how to leverage source-domain data when its dynamics do not match the target domain. Instead of following the common approach of filtering or penalizing mismatched data, the authors propose a more proactive strategy: directly modifying source transitions to better align with the target domain.

Specifically, the proposed method, Selective Transition Correction (STC), first uses an inverse policy model to infer target-consistent actions for source transitions and adjusts rewards accordingly via a learned reward model. To mitigate errors from imperfect models (especially under limited target data), they introduce a forward dynamics model to selectively accept only those corrected transitions that better match target dynamics. The paper also provides a theoretical analysis showing that corrected transitions can reduce dynamics mismatch and bound performance gaps, and demonstrates empirical gains across various benchmark settings.

The main strength of the work is its clear conceptual shift from discarding source data to transforming it, supported by a well-designed combination of inverse and forward models and solid empirical validation. However, the method introduces additional complexity by requiring multiple auxiliary models, and its performance depends heavily on their accuracy. Some theoretical assumptions may be restrictive, and parts of the approach (such as reward correction and selection thresholds) are heuristic and require tuning.

**Audience:**

Yes

**Audience Explanation:**

The gains of this work are mainly from preprocessing, not learning. This means that the improvements come mostly from better data, not from fundamentally better policy optimization. So the contribution is somewhat pipeline-level, not algorithmically deep in the RL core. However, individuals working on preprocessing might be interested in this paper.

**Claims And Evidence:**

No

**Claims Explanation:**

My major concern is the heavy reliance on learned models. Specifically, the method depends on three learned components: (1) inverse policy model (for action correction); (2) reward model (for reward adjustment); (3) forward dynamics model (for filtering).
And the method may fail if any of these are inaccurate, which is likely given the limited target data.
Although the selection mechanism mitigates this, it sacrifices the potential of the proposed method (i.e., the less data selected, the less useful the proposed method). In addition, this approach presents a contradiction: if the goal is to solve the problem of limited target data, how can this method learn reliable inverse models from these limited target data? How much target data is required to learn an effective model?

Other weaknesses include (1) the introduction of additional parameters that require tuning; (2) limited robustness to large domain shifts as the approach assumes that: source and target dynamics are not too far apart, and corrected actions can meaningfully align transitions; (3) and strong/optimistic theoretical assumptions that may not hold in real-world settings. But these are relatively minor.

**Requested Changes:**

As mentioned above, my major concern is the heavy reliance on learned models that creates a contradiction: if the goal is to solve the problem of limited target data, how can this method learn reliable inverse models from these limited target data? My recommendation would depend mostly on how the authors address this concern.

---

> ### Author Response · Authors · 2026-05-02
> **Author Responses to Reviewer pxSH (part 1)**
>
> We sincerely thank the reviewer for the valuable comments. Below, we provide point-by-point responses to the reviewer’s comments and describe the corresponding revisions made to the manuscript.
>
>
> **Concern 1: If the goal is to solve the problem of limited target data, how can this method learn reliable inverse models from these limited target data?**
>
> We thank the reviewer for raising this important question. We address it from two complementary angles: the feasibility of model learning and the robustness of the framework to imperfect models.
>
> **On model learning feasibility.** In our work, limited target data refers to a realistic low-data regime (5k transitions, consistent with prior work such as OTDF [1]) in which learning a high-quality policy directly from target data is difficult, rather than a regime in which learning auxiliary models is infeasible. In our implementation, the inverse model is pretrained for 50k steps before policy learning, and its training loss steadily converges to a small magnitude ($\sim10^{-4}$), indicating reliable data fitting at this scale. This is consistent with prior findings: for instance, the pen-human and door-human datasets in D4RL [2] (~5k–6k transitions) have successfully supported model learning in model-based offline RL methods [3,4] on the more complex Adroit tasks. Since the MuJoCo locomotion tasks in our work are comparatively simpler, 5k transitions are sufficient to train an inverse model with reasonable accuracy.
>
> **On robustness to imperfect models.** Crucially, the inverse model in our framework is not required to perfectly recover ground-truth actions. It serves as a proposal model that generates candidate action corrections, and its outputs are subsequently validated by a forward dynamics model: a correction is applied only when it improves transition consistency with the target domain. This selective mechanism ensures that even an imperfect inverse model contributes positively, as implausible proposals are filtered out while beneficial ones are retained.
>
> Empirically, baseline methods that rely solely on target data suffer from significant performance degradation in this regime, whereas STC remains stable across tasks, confirming that the learned inverse model provides effective and reliable correction signals even under limited data.
>
>
>
> **Concern 2: The method heavily relies on three learned components, and may fail if any of them is inaccurate.**
>
>
>
> We thank the reviewer for this insightful concern. We agree that the performance of STC depends on learned components, but the method is designed to be robust to moderate inaccuracies in any individual component, rather than requiring all components to be highly accurate in isolation.
>
> The key reason is that the three components play different and complementary roles. The inverse model proposes candidate action corrections, the forward model evaluates whether a proposed correction improves consistency with the target dynamics, and the reward model is only used for a first-order reward adjustment. As a result, STC does not behave like a purely sequential pipeline in which errors from each model are directly accumulated.
>
> In particular, the selective correction mechanism makes the overall procedure conservative: a correction is applied only when the forward model indicates that it better matches the target transition than the original source action. This introduces a mutual check between the inverse and forward models, so an inaccurate proposal is unlikely to be accepted unless it also improves forward consistency. In addition, the learned reward model is only used to adjust source rewards via a first-order approximation, which further mitigates the risk of introducing large estimation errors.
>
> Finally, empirical results further support this claim. Despite using only 5k target-domain transitions, the learned models provide effective correction signals. As shown in Figure 2 in the paper, the distribution of corrected source transitions aligns closely with that of the target domain. Moreover, Table 1 in the paper demonstrates that STC significantly outperforms all baselines, indicating that the method remains effective even when trained under limited data and with imperfect learned components.

---

> > ### Author Response · Authors · 2026-05-02
> > **Author Responses to Reviewer pxSH (part 2)**
> >
> > **Concern 3:  the selection mechanism sacrifices the potential of the method, as selecting fewer data reduces its overall usefulness.**
> >
> > We thank the reviewer for this thoughtful concern. We agree that the selective mechanism reduces the number of corrected transitions, but we would like to emphasize that the usefulness of correction depends on its quality, not simply its coverage. Correcting more transitions is not necessarily beneficial if some of those corrections are unreliable and introduce bias into policy learning.
> >
> > As shown in the ablation study in Table 1, applying action and reward correction to all source transitions does not consistently improve performance. While it helps on some tasks, it causes clear degradation on others. In contrast, incorporating the selective mechanism leads to more stable and consistently stronger results across tasks. This suggests that unrestricted correction does not increase the potential of the method in practice; instead, it can reduce effectiveness by introducing harmful corrections.
> >
> > The reason is that some source transitions may lie outside the reliable generalization region of the learned correction models. For such transitions, the proposed action or reward correction may be inaccurate, and forcing correction on them can hurt downstream learning. The selective mechanism is therefore introduced to retain high-confidence corrections while filtering out potentially harmful ones, making the overall adaptation process more conservative and robust.
> >
> > At the same time, the mechanism does not discard correction to the point of losing usefulness. As shown in Table 2, a substantial fraction of transitions are still corrected across tasks, indicating that correction remains an important contributor to performance. Overall, the selective mechanism provides a better trade-off between correction coverage and correction reliability, which leads to stronger and more consistent results in practice.
> >
> >
> >
> > | Type    | Source                     | Target | w/o selection | STC (ours) |
> > |---------|----------------------------|--------|----------------|-------------|
> > | Gravity | walker2d-medium            | medium | 39.0±4.1    | **41.6±4.0** |
> > | Gravity | halfcheetah-medium         | medium | 27.6±5.6     | **42.4±5.3** |
> > | Gravity | hopper-medium              | medium | 37.9±5.5     | **43.4±6.1** |
> > | Morph   | halfcheetah-medium   | medium | 4.2±8.4      | **19.5±2.2** |
> > | Morph   | walker2d-medium        | medium | **59.7±8.3** | 56.7±8.1 |
> > | Morph   | hopper-medium         | medium | 42.8±7.7 | **43.1±23.9** |
> > | **Total Score**     |                            |      | 211.2         | **246.80** |
> >
> > Table 1: Ablation study on the effect of the selective mechanism for source transition correction.
> >
> >
> >
> > | Environment                   | Correction percentage |
> > |------------------------------|------------------------|
> > | hopper-gravity-shift         | 89.84\%                |
> > | ant-gravity-shift            | 96.88\%                |
> > | walker2d-gravity-shift       | 25.78\%                |
> > | halfcheetah-gravity-shift    | 31.25\%                |
> > | hopper-morphology-shift      | 91.41\%                |
> > | ant-morphology-shift         | 92.19\%                |
> > | walker2d-morphology-shift    | 93.75\%                |
> > | halfcheetah-morphology-shift | 42.19\%                |
> >
> > Table 2. The percentage of source-domain transitions corrected.
> >
> >
> >
> >
> >
> > **Concern 4: The method introduces additional hyperparameters that require tuning.**
> >
> >
> > We thank the reviewer for this observation. Our method introduces three additional hyperparameters: the correction threshold $\lambda$, the reward gradient coefficient $\alpha$, and the Q-weighted loss coefficient $\beta$. As with most RL methods, these parameters can affect performance if chosen poorly, and tuning them within a reasonable range is standard practice. We also note that prior cross-domain offline RL methods, such as IGDF [5] and OTDF [1], similarly require hyperparameter tuning to achieve their best performance. In our experiments, we follow the settings and tuning protocols in the original papers for all baselines to ensure a fair comparison.
> >
> > In practice, however, the tuning required by our method is limited. We fix
> > $\alpha=0.5$  across all tasks, as this value performs consistently well. For
> > $\lambda$, we consider only two candidate values, 1.0 and 5.0, both of which yield strong performance across environments. Similarly, for $\beta$, we use only a small set of values, 0.5 and 5.0, with 5.0 working well in most cases. The fact that strong results can be obtained from such a restricted search space suggests that our method does not rely on extensive hyperparameter tuning.

---

> > > ### Author Response · Authors · 2026-05-02
> > > **Author Responses to Reviewer pxSH (part 3)**
> > >
> > > **Concern 5: limited robustness to large domain shifts.**
> > >
> > > Thanks for the comment. We agree that handling large domain shifts remains challenging. When the domain shift becomes too large, the effectiveness of our approach can be limited. However, we would like to first clarify that the assumption of moderate domain shift is not a strong restriction in practice.
> > > Our method is not aimed at arbitrarily large domain shifts, but at a more practical and commonly encountered setting: similar environments with mismatched dynamics. This scenario arises frequently in real-world applications, such as transferring policies between simulation and real-world robotic systems, where the state and action spaces are shared but the dynamics differ. For such settings,  our method STC can be effective and practically valuable.
> > >
> > > Regarding the concern that corrected actions may not meaningfully align transitions: our selective correction mechanism directly relaxes this assumption. Rather than requiring all proposed corrections to be accurate, we apply a correction only when the forward model confirms that it reduces the prediction error under target dynamics. Corrections that fail this check are discarded. This means the method does not rely on every correction being meaningful, as the selection process automatically identifies and retains only those corrections that are valid.
> > >
> > >
> > > **Concern 6: The theoretical analysis relies on strong or optimistic assumptions that may not hold in real-world settings.**
> > >
> > > We thank the reviewer for this concern. We acknowledge that the assumptions underlying our theoretical analysis are simplifications that may not strictly hold in practice. However, we emphasize that the purpose of the theoretical analysis is not to provide guarantees under general real-world conditions, but rather to offer a principled justification for the core design choice of transition correction. By working in a tractable, idealized setting, we are able to formally characterize when and why correcting source transitions can improve alignment with the target dynamics, which provides a clear theoretical motivation for our method.
> > >
> > > We would also note that this is consistent with standard practice in offline RL theory, where simplifying assumptions are commonly adopted to make the analysis tractable while still yielding meaningful insights. The empirical results in Table 1 and Figure 2 further complement the theoretical analysis by demonstrating that the benefits of transition correction hold robustly across a range of practical settings, even when the idealized assumptions are not fully satisfied. We will add a clarification in the revised manuscript to make the scope and intent of the theoretical analysis more explicit.
> > >
> > >
> > > **Changes Made in the Revised Manuscript**
> > >
> > > Based on the concerns raised above, we have made corresponding revisions to the manuscript to strengthen the clarity and completeness of the paper. All modified parts are highlighted in red, and the detailed descriptions of each change are provided below.
> > >
> > > **Change 1: Clarification on the feasibility of learning an inverse model under limited target data.**
> > >
> > > We add clarification in the Section 1 (Introduction) and Section 4.3 (Selective Correction Mechanism) of the revised manuscript, we clarify that limited target data refers to a realistic low-data regime of approximately 5k transitions, consistent with prior work such as OTDF [1]. This scale is not insufficient for learning a high-quality policy directly from target data alone, but sufficient for learning auxiliary models (e.g., inverse and forward dynamics models). This distinction is important for correctly interpreting the motivation and scope of our method. And the training loss in practice can indicate reliable data fitting at this scale.
> > >
> > > **Change 2: Empirical analysis of the selective correction mechanism**
> > >
> > > In response to Concern 3, we have added empirical evidence to further validate the role and effectiveness of the selective correction mechanism:
> > >
> > > In Section 5.5, we add an ablation study comparing STC with and without the selective correction mechanism. The results demonstrate that removing the selection step leads to a consistent performance drop across tasks, confirming that ensuring correction quality is more critical than maximizing the number of corrected transitions.
> > >
> > > In Appendix E.3, we further report the Correction Rate of Source-Domain Transitions across different tasks. The results show that a substantial proportion of source transitions are successfully retained after selection, indicating that the mechanism does not excessively filter out data in practice, and the corrected transitions provide sufficient training signal for effective policy learning.

---

> ### Author Response · Authors · 2026-05-02
> **Author Responses to Reviewer pxSH (part 4)**
>
> **Change 3: Clarification on the assumptions of theoretical analysis**
>
> We further elaborate in the theoretical analysis section of the revised manuscript on the idealized assumptions and their gap to real-world settings, and clarify how these assumptions guide the design of our method. The revised parts are highlighted in red.
>
> **Change 4: Add discussion on robustness to imperfect learned components**
>
> We have added a discussion on the robustness of STC to imperfect learned components in Appendix F of the revised manuscript, clarifying the complementary roles of different models and how the selective correction mechanism mitigates error propagation. The corresponding modifications are highlighted in red.
>
>
> We hope that the above responses and revisions can resolve the concerns. If there is still something unclear, please let us know!
>
> [1] Cross-domain offline policy adaptation with optimal transport and dataset constraint.
>
> [2] D4RL: Datasets for Deep Data-Driven Reinforcement Learning.
>
> [3] Model-based offline planning with trajectory pruning.
>
> [4] Model-Bellman inconsistency for model-based offline reinforcement learning.
>
> [5] Contrastive representation for data filtering in cross-domain offline reinforcement learning.

---

### Review · Reviewer_SV7t · 2026-04-23

**Summary Of Contributions:**

The paper proposes Selective Transition Correction (STC) for cross-domain offline RL with dynamics mismatch. Instead of only filtering or penalizing source-domain data, it edits source transitions to better match the target domain: an inverse model corrects actions, a reward model adjusts rewards, and a forward model decides whether to keep the corrected transition. The paper also provides supporting theory and shows strong empirical gains on MuJoCo tasks with gravity, friction, and morphology shifts.

**Audience:**

Yes

**Audience Explanation:**

The topic is interesting and important.

**Broader Impact Concerns:**

No concern.

**Claims And Evidence:**

Yes

**Claims Explanation:**

The main claims are supported by solid empirical evidence: the paper evaluates on multiple tasks and dynamics shifts, compares against several relevant baselines, and reports consistently strong gains. The paper also includes ablations and visualizations that support the correction mechanism, and the theory is aligned with the method’s intuition. The main limitation is that the theoretical results rely on strong assumptions and the baseline set could be broader, but overall the evidence is convincing for the paper’s central claims.

**Requested Changes:**

1. Add a clear target-only baseline with the same offline RL backbone, to isolate the value of using corrected source data.
2. Strengthen the core ablation with always-correct and no-selection / inverse-only variants, since the main contribution is selective correction

---

> ### Author Response · Authors · 2026-05-02
> **Author Responses to Reviewer SV7t (part 1)**
>
> We sincerely thank the reviewer for the thoughtful comments. We have carefully examined the concerns raised and the suggested revisions, and have accordingly conducted additional experiments and updated the manuscript. Detailed responses are provided below.
>
> **Concern 1:theoretical results rely on strong assumptions.**
>
> We thank the reviewer for this important comment. We agree that the theoretical analysis relies on simplifying assumptions, which may not fully hold in practice. The intention of the theory is to provide intuition and partial justification for the proposed design, rather than to precisely capture all complexities encountered in empirical settings. In this sense, the theoretical results should be interpreted as offering insight into the mechanism of the method under idealized conditions, while our experiments in simulated environments demonstrate its effectiveness under more realistic dynamics and data conditions beyond the scope of the theory.
>
> We would also note that this is consistent with standard practice in offline RL methods (such as OTDF [1] and IGDF [2]), where simplifying assumptions are commonly adopted to make the analysis tractable while still yielding meaningful insights.
>
> **Changes Made in the Revised Manuscript**
>
> Based on the reviewer’s concerns and suggestions, we have revised the manuscript accordingly. The changes are detailed as follows:
>
>
> **Change 1: Add a clear target-only baseline with the same offline RL backbone, to isolate the value of using corrected source data.**
>
>
> We thank the reviewer for this helpful suggestion. We agree that introducing a target-only baseline using the same offline RL backbone is important for isolating the contribution of corrected source transitions. In the revised manuscript, we have added this comparison to explicitly evaluate whether the performance gains come from incorporating corrected source data.
>
> As shown in the results of Table 1, the target-only baseline performs significantly worse than STC across most tasks, with particularly large gaps in *halfcheetah* and *walker2d* tasks. In addition, in some tasks, training solely on the target dataset even leads to policy collapse due to the limited size of the target data. Overall, STC achieves a substantially higher total score (655.2 vs. 457.2), demonstrating that leveraging corrected source transitions provides clear and consistent benefits beyond training solely on target data.
>
> We have added this ablation in Appendix E.6 of the revised manuscript to strengthen the empirical validation of our method. The corresponding modifications have been highlighted in red in the revised submission.
>
> | Type    | Source           | Target  | Target-only  | STC (ours)   |
> |---------|------------------|---------|--------------|--------------|
> | Gravity | halfcheetah-med  | medium  | 0.6±0.5      | 42.4±5.3     |
> | Gravity | halfcheetah-med  | med-e   | 0.0±0.1      | 37.3±1.5     |
> | Gravity | hopper-med       | medium  | 46.2±3.7     | 43.4±6.1     |
> | Gravity | hopper-med       | med-e   | 18.6±13.1    | 23.4±6.4     |
> | Gravity | walker2d-med     | medium  | 28.5±7.4     | 41.6±4.0     |
> | Gravity | walker2d-med     | med-e   | 17.9±11.4    | 36.9±6.5     |
> | Gravity | ant-med          | medium  | 32.0±7.6     | 42.6±8.4     |
> | Gravity | ant-med          | med-e   | 16.9±4.5     | 23.2±6.1     |
> | Morph   | halfcheetah-med  | medium  | 12.3±3.7     | 19.5±2.2     |
> | Morph   | halfcheetah-med  | expert  | -0.7±0.6     | 7.7±1.2      |
> | Morph   | hopper-med       | medium  | 42.5±17.3    | 43.1±23.9    |
> | Morph   | hopper-med       | expert  | 62.3±28.0    | 53.2±50.3    |
> | Morph   | walker2d-med     | medium  | 43.1±11.5    | 56.7±8.1     |
> | Morph   | walker2d-med     | expert  | 9.9±4.1      | 32.9±7.3     |
> | Morph   | ant-med          | medium  | 76.8±2.8     | 77.2±2.8     |
> | Morph   | ant-med          | expert  | 50.3±12.2    | 74.1±21.2    |
> | **Total Score** |          |         | 457.2        | 655.2        |
>
> Table 1: Comparison between target-only baseline and STC.

---

> ### Author Response · Authors · 2026-05-02
> **Author Responses to Reviewer SV7t (part 2)**
>
> **Change 2: Strengthen the core ablation with always-correct and no-selection / inverse-only variants, since the main contribution is selective correction.**
>
> We thank the reviewer for the helpful suggestion. To better isolate the effect of the selective correction mechanism, we add an ablation comparing the full method with a variant that removes the selection step and applies inverse correction to all source transitions, i.e., an always-correct setting.
>
> As shown in Table 2, removing the selective mechanism leads to performance degradation on most tasks, with particularly notable drops on halfcheetah-gravity, halfcheetah-morph, and hopper-gravity. This suggests that directly applying inverse corrections to all transitions is not always reliable. In practice, some source transitions may lie outside the reliable generalization region of the learned inverse model, and blindly correcting them can introduce misaligned actions that ultimately harm policy learning.
>
> By contrast, the selective mechanism keeps only those corrections that improve consistency under the forward model, leading to more stable and generally better performance. Although the gap is small in a few cases, the overall trend is clear: selective correction improves robustness by preventing low-quality corrections from being applied indiscriminately. Overall, the results demonstrate that the selective mechanism is a key component for achieving robust gains.
>
> The corresponding ablation results have been added in Section 5.5 of the revised manuscript and are highlighted in red.
>
>
>
> | Type    | Source                     | Target | w/o selection | STC (ours) |
> |---------|----------------------------|--------|----------------|-------------|
> | Gravity | walker2d-medium            | medium | 39.0±4.1    | **41.6±4.0** |
> | Gravity | halfcheetah-medium         | medium | 27.6±5.6     | **42.4±5.3** |
> | Gravity | hopper-medium              | medium | 37.9±5.5     | **43.4±6.1** |
> | Morph   | halfcheetah-medium   | medium | 4.2±8.4      | **19.5±2.2** |
> | Morph   | walker2d-medium        | medium | **59.7±8.3** | 56.7±8.1 |
> | Morph   | hopper-medium         | medium | 42.8±7.7 | **43.1±23.9** |
> | **Total Score**     |                            |      | 211.2         | **246.80** |
>
> Table 2: Ablation study on the effect of the selective mechanism for source transition correction.
>
> **Change 3: Clarification on the assumptions of theoretical analysis**
>
> We further elaborate in the theoretical analysis section of the revised manuscript on the idealized assumptions and their gap to real-world settings, and clarify how these assumptions guide the design of our method. The revised parts are highlighted in red.
>
>
> We hope that the above responses address the reviewer’s concerns. Please let us know if further clarification is needed.
>
> [1] Cross-domain offline policy adaptation with optimal transport and dataset constraint.
>
> [2] Contrastive representation for data filtering in cross-domain offline reinforcement learning.